# Colonization of macroalgal deposits by estuarine nematodes through air and potential for rafting inside algal structures

**Bartelijntje Buys**[1], **Sofie Derycke**[1,2¤], **Nele De Meester**[1,2], **Tom Moens** [1]*

**1** Department of Biology, Marine Biology Lab, Ghent University, Ghent, Belgium, **2** Center for Molecular Phylogeny and Evolution, Ghent University, Ghent, Belgium

¤ Current address: Aquatic Environment and Quality, Research Institute for Agriculture, Fisheries and Food, Ostend, Belgium
* tom.moens@ugent.be

**Data Availability Statement:** All data of the study are available in the Integrated Marine Information System (IMIS) database (VLIZ): https://doi.org/10.14284/455.

## Abstract

Dispersal is an important life-history trait. In marine meiofauna, and particularly in nematodes, dispersal is generally considered to be mainly passive, i.e. through transport with water currents and bedload transport. Because nematodes have no larval dispersal stage and have a poor swimming ability, their per capita dispersal capacity is expected to be limited. Nevertheless, many marine nematode genera and even species have near-cosmopolitan distributions, and at much smaller spatial scales, can rapidly colonise new habitat patches. Here we demonstrate that certain marine nematodes, like the morphospecies *Litoditis marina*, can live inside macroalgal structures such as receptacula and–to a lesser extent–floating bladders, which may allow them to raft over large distances with drifting macroalgae. We also demonstrate for the first time that these nematodes can colonize new habitat patches, such as newly deposited macroalgal wrack in the intertidal, not only through seawater but also through air. Our experimental set-up demonstrates that this aerial transport is probably the result of hitchhiking on vectors such as insects, which visit, and move between, the patches of deposited algae. Transport by wind, which has been observed for terrestrial nematodes and freshwater zooplankton, could not be demonstrated. These results can be important for our understanding of both large-scale geographic distribution patterns and of the small-scale colonization dynamics of habitat patches by marine nematodes.

## Introduction

Dispersal allows a species to escape unfavorable conditions (e.g. competition, resource depletion, local disturbance events) and enhances gene flow [1–3]. Marine dispersal has long been considered 'unlimited' because seas, oceans and estuaries seem highly interconnected environments with only few barriers, but see [4–6]. Marine and estuarine organisms mainly disperse through active swimming (e.g. fish) or through drifting planktonic larvae (e.g. many macrobenthos species). Still, even species with limited swimming capacities or without pelagic larvae

**Funding:** Financial support for this work was obtained from the Flemish Science Fund FWO through project nr. G038715N, awarded to T.M. and S.D. The funders had no role in study design, data collection and analysis, decision to publish, or preparation of the manuscript. https://www.fwo.be/
.

**Competing interests:** The authors have declared that no competing interests exist.

can disperse over considerable distances: Bedload transport through water currents may be one of the most important and consistent dispersal mechanisms for small benthic organisms lacking pelagic larval stages, particularly in coastal zones [7–10]. In addition, drifting (floating passively, e.g. through larva propagules), rafting (attached to a floating object), hitchhiking (attached to a mobile object or animal), hopping (going from one suitable habitat/substratum to another) or creeping (over substrates/habitats which are similar over a considerable distance) can facilitate dispersal over variable distances [11].

Members of the marine and estuarine meiofauna, a taxonomically and functionally diverse group of small organisms (passing through a 1-mm mesh but being retained on a 38-μm mesh), typically have limited active dispersal capacities and lack a pelagic larval stage. Nematodes are usually by far the numerically dominant meiofaunal taxon in soft sediments, and among the most abundant on algae, seagrasses and salt marsh vegetation [12, 13]. Their dispersal is considered to be mostly passive, mediated by bedload transport, by drifting in sea water after erosion from sediments or phytal substrata, or through 'rafting' on vectors such as drifting macroalgae or sea turtles [7, 10, 14–18]. Nevertheless, some nematodes may actively enter the water column, thus facilitating their dispersal through water currents and bedload transport [7, 19–21], and they appear capable of choosing sites where they will settle again [21, 22].

Recent studies suggest that effective meiofaunal dispersal nevertheless may be restricted in space, even at scales of (tens of) kilometers [23–27]. This seems at odds with their often cosmopolitan distributions [11, 17], but can be explained by occasional long-distance dispersal [28, 29] and the cumulative effect of relatively short-distance but frequent dispersal events through bedload transport [7–10], leading to a wide geographic distribution of species. In addition, species' distributions may sometimes be overestimated, specifically when morphospecies comprise a complex of morphologically indistinguishable cryptic species [30, 31] with partly disparate distribution ranges. Both the presence of substantial cryptic diversity [31] and of occasional long-distance dispersal events [28] have been demonstrated in the marine nematode species complex *Litoditis marina*, a common inhabitant of decaying and standing live macroalgae in coastal and estuarine habitats [32, 33]. *Litoditis marina* also exhibits strong colonization-extinction dynamics, the presence of nearby source populations greatly facilitating rapid colonization of new patches [34]. Indeed, the macroalgal habitat is often ephemeral and patchily distributed. *Litoditis marina* has a short generation time and a high reproductive output [35, 36], implying that if it manages to reach new patches, its dispersal has a high probability of being effective.

*Litoditis marina* has evolved within a largely terrestrial nematode clade [33, 37] where insects and other arthropods are often used as hosts or vectors of several parasitic, commensal and free-living representatives [32, 33, 38]. Like *L. marina*, many free-living soil Rhabditidae live in patchily distributed, nutrient-rich environments, and quite a few of those species disperse through air, either by hitchhiking on arthropods [38] or carried by wind and/or rain [39]. The possibility of hitchhiking on arthropod vectors has received little attention in marine nematodes (but see [33, 40]), and the few papers which have investigated aerial nematode dispersal without the help of a vector [41] have invariably focused on terrestrial or freshwater species [39, 41–45]. Nevertheless, wind dispersal could potentially also transport nematodes from exposed littoral environments such as sandy beaches, salt marshes and intertidal algal holdfasts.

Another possibility for marine nematode dispersal (from short to long distances) is through rafting on macroalgae. Even mild currents can wipe a substantial fraction of nematodes off plant or macroalgal surfaces [21, 46], but some free-living marine nematodes may anchor themselves to algal substrata through mucus secretions produced by the caudal glands [47], or find shelter in between or perhaps even inside plant or algal structures [48–51].

In this study, we investigated nematode dispersal through air, as well as the possibility that nematodes can be transported inside algal structures, using a set of dedicated experiments in a salt marsh in a macrotidal estuary. The latter behaviour may benefit both short- and long-distance dispersal via rafting, as nematodes are unlikely to be wiped off by seawater currents when they are protected within these algal structures.

## Materials and methods

Three experiments were performed; in the first two, we investigated marine nematode colonization of algae through the air, while in the third one we explored the ability of nematodes to use macroalgal structures as rafts. For all experiments, algae were collected from the eastern edge of the Paulina salt marsh (51.36˚N, 3.69˚E) in the polyhaline reach of the Schelde Estuary (The Netherlands). This marsh is bordered by a dyke and by tidal mudflats. Algal stands (mainly *Fucus* sp.) occur on a few deposits of stones mainly at the basis of the dyke and on an isolated breaker near (ca. 50 m distance from) the downstream (western) border of the marsh, whereas algal wrack is irregularly deposited in small patches along the edges of the marsh or inside small creeks and gullies and amidst vegetation. For a detailed geomorphological description of this marsh, see [52]. Field samples were collected at the Paulina polder tidal flat in collaboration with NIOZ (Netherlands Institute of Sea Research), who provided the necessary permit for field sampling, issued by the "Provincie Zeeland, The Netherlands: Directie Ruimte, Milieu en Water".

We did not systematically identify nematodes in our experiments, but screened for the presence of *L. marina*, as nematodes of this species complex are among the dominant inhabitants of decaying macroalgae, and–through their close phylogenetic relationship with terrestrial nematode species that use arthropod vectors for their dispersal–would be top candidates for vector-borne dispersal.

### Can marine nematodes disperse through air?

**Experimental set up.**   We performed a first field experiment to test if nematodes can disperse between patches of algae through tidal currents, air or both, and whether any airborne dispersal would be through hitchhiking on flying vectors. This experiment was performed in September 2013 and lasted one week. Average minimum and maximum field temperatures were 6.6 and 13˚C, respectively. Precipitation rate averaged 9.9 mm/day and mean wind velocity was 3 Bft, with wind coming from the north on four sampling days and from the southwest on the remaining three days.

The algae collected at Paulina (*Fucus vesicolosus* with some very sparse 'traces' of *F. spiralis)* were defaunated by immersion in tap water for 12 h, followed by drying at 60˚C for 54 h to kill all nematodes, including resting stages and eggs. We checked the efficiency of the defaunation procedure by incubating several pieces of defaunated algae (up to a total of 100 g dry weight) on a sloppy marine agar medium under the same conditions and for the same duration as the algae from the field experiments (see below). No living nematodes were ever observed in these controls, confirming that our defaunation method was 100% effective.

Subsequently, 200-g portions of defaunated algae were rehydrated in artificial seawater (ASW) and added to waterproof (using silicon lining) wooden boxes (H 25 x L 23 x W 23 cm) with a slightly pitched shelter (positioned 10 cm above the edge of the box) (Fig 1). The shelter mainly served to avoid impacts of rain and sunlight (Fig 1).

The boxes were moored onto 3-m high metal poles in the field in between the marsh vegetation, ca. 1.36 m above the maximum tide level (calculated from the sum of the aboveground height of the poles (ca. 1.75 m) and a field elevation of ca. 2 m above sea level, minus the

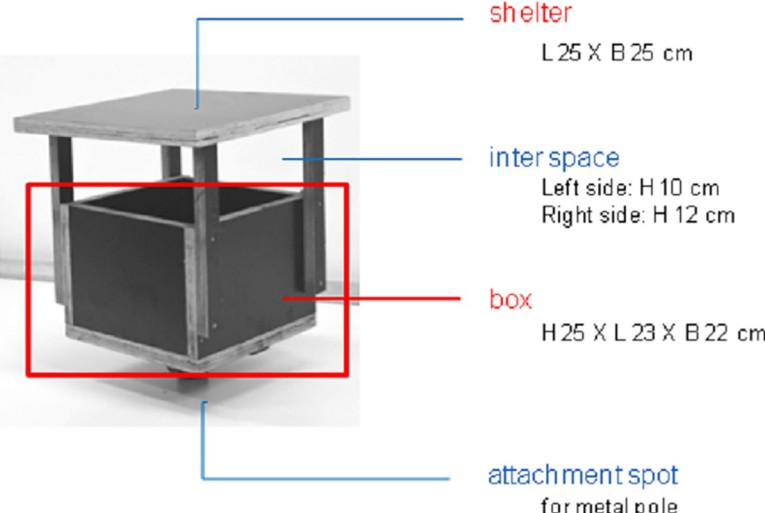

**Fig 1. Boxes mounted upon poles to investigate marine nematode dispersal through air in the field.** Wooden boxes (H 25 x L 23 x W 23 cm) with shelter, containing 200 g defaunated algae, were placed on metal poles ca. 1.75 m above sediment level amidst salt marsh vegetation.

maximal tidal height of 2.39 m (https://getij.rws.nl/)). There were two types of boxes: (1) open boxes were not covered except by the wooden shelter and could be visited freely by insects and other arthropods; and (2) boxes covered with a gauze with a mesh size of 200 μm prevented entry of insects, spiders and other potential vectoring fauna, but allowed entry of nematodes transported by wind or dropping off from vectors landing on top of the gauze [39, 44, 45]. 100 ml ASW was added to each box to slow down possible dehydration resulting from environmental factors such as wind. We haphazardly placed the 4 open and 4 gauzed boxes in the field within a rectangular section of ca. 14 m x 6 m of the high marsh [52], parallel with the water line.

We sampled the boxes by transferring the algae to clean buckets. In the laboratory, the algae of each box were spread out in 6 9-cm diameter Petri dishes on a sloppy (0.7%) marine agar medium, composed of bacto and nutrient agar in a ratio of 10 to 1 and with a salinity of 25 [53]; they were incubated at 20°C in the dark. *Litoditis marina*, but also many other marine nematode species, tend to move into the agar within a few days after incubation [53], where they can be easily observed and counted with a Leitz Dialux stereomicroscope (14x-60x magnification). The agar plates were checked after 4 days, which corresponds to one complete generation time of *L. marina* [36, 54] under the incubation conditions used here. Because not all nematodes will readily leave the algal substratum, we increased the likelihood of observing the presence of living nematodes (including eggs glued to the algal surface [47] or dauer larvae) in the agar by incubating the plates up to 14 days, with counts after 7 and 14 days. In addition, we sampled the artificial seawater which was present inside the boxes and screened it for the presence and abundance of nematodes upon return to the laboratory. In this first exploratory experiment, we did not identify any nematodes encountered on the algae or in the associated seawater. In fact, no living nematodes invaded the agar plates, and any nematodes retrieved from the seawater in the boxes were dead and in a poor state, hampering their identification.

**Data analysis.**   Because no nematodes were observed on the agar plates for any of the treatments, only data of nematodes observed in the water inside the boxes at the end of the field incubations were used for further analysis. An independent t-test was performed in R to compare the average numbers of nematodes in the water in the open and gauzed boxes. Assumptions of normality of data and heterogeneity of variances were checked on our log-

transformed dataset using, respectively, a Shapiro–Wilk test and a Levene test in the R software program (version 3.1.0, R developmental Core Team 2008).

## Are decomposing macroalgae colonized by nematodes through air or through the water column?

**Experimental set up.** To compare nematode colonization of algae through the air with colonization through seawater, we repeated the above-described experiment in a modified design in April 2014 for a duration of two weeks. The average air temperature was 10.5˚C with almost no rain (0.03 mm/day) and a mean wind velocity of 4.1 Bft, predominantly coming from southerly directions. We used the presence of a small cliff, abruptly 'separating' the middle from the lower marsh [52], to position boxes with defaunated algae which could only become colonized by nematodes through air and boxes which could also become flooded or at least 'splashed' at high tide at a distance of < 20 m, thereby greatly limiting the possible confounding effect of a different environment (with a different vegetation, candidate vectors etc. . .). Four open boxes with rehydrated defaunated algae were mounted onto 3-m high poles and placed ca. 1.75 m above the maximal tide level, while four others were positioned at ca. 0.27 m below the average tidal level. These treatments are referred to as 'air only' and 'air + seawater', respectively. Both sets of boxes were positioned parallel to the water line.

To counteract drying of the algae, we added 250 mL of sterile seawater to all the boxes at the beginning of the experiment and again after one week of incubation in the field. We sampled all boxes after 14 days, following the same procedure as in the first experiment, but determination of presence and quantification of nematodes were done differently. In short, in the lab, algae were thoroughly rinsed with tap water (15 min/box) over a 38-μm sieve to collect all nematodes present on the algae. Any water (ASW in the higher 'air only' boxes, and natural tidal field seawater in the 'air + seawater' boxes) was also poured over the same sieve. The nematodes were collected in ASW with a salinity of 15, and stored at 4˚C before counting. Nematode samples were mixed using a magnetic stirrer, and four subsamples of 700 μl (which amounts to ca 5% of the total sample volume of 60 ml) were counted; the abundance of nematodes in those aliquots was back-calculated to the total sample volume (60 ml), so that we had a good approximation of the total number of nematodes inside each box at the time of sampling. Because a systematic identification of all the nematodes in the boxes was beyond the scope of our experiment, we randomly screened ca 100 nematodes per sample to assess whether *Litoditis* sp. were present. For this screening, nematodes were transferred through anhydrous glycerin and mounted in permanent microscopic slides [55]. Reliable identification of many nematodes was hampered due to a large number of juveniles and/or to morphological degradation / damage due to harsh field conditions.

**Data analysis.** Differences between nematode abundances inside the 'air only' and the 'air + seawater' boxes were assessed using a t-test on the log transformed abundance data.

## Can macroalgae serve as a shelter for nematodes?

**Experimental set up.** In this experiment we investigated whether nematodes can reside inside macroalgal structures like the floating bladders and receptacula (i.e. bladder-like reproductive organs) (Fig 2). We haphazardly collected 17 complete thalli of the brown alga *F. vesiculosus* growing on stones near the upper edge of the Paulina salt marsh on September 10, 2013 at low tide. In the laboratory, we separated 14 floating bladders of ca. 1.0 cm$^2$ and 20 receptacula of ca. 1.5 cm$^2$ using clean scissors. Each structure was thoroughly rinsed with tap water; the tap water causes an osmotic shock which temporarily 'sedates' the nematodes, facilitating their removal from surfaces [56]. We collected the tap water and sieved it over a 38-μm

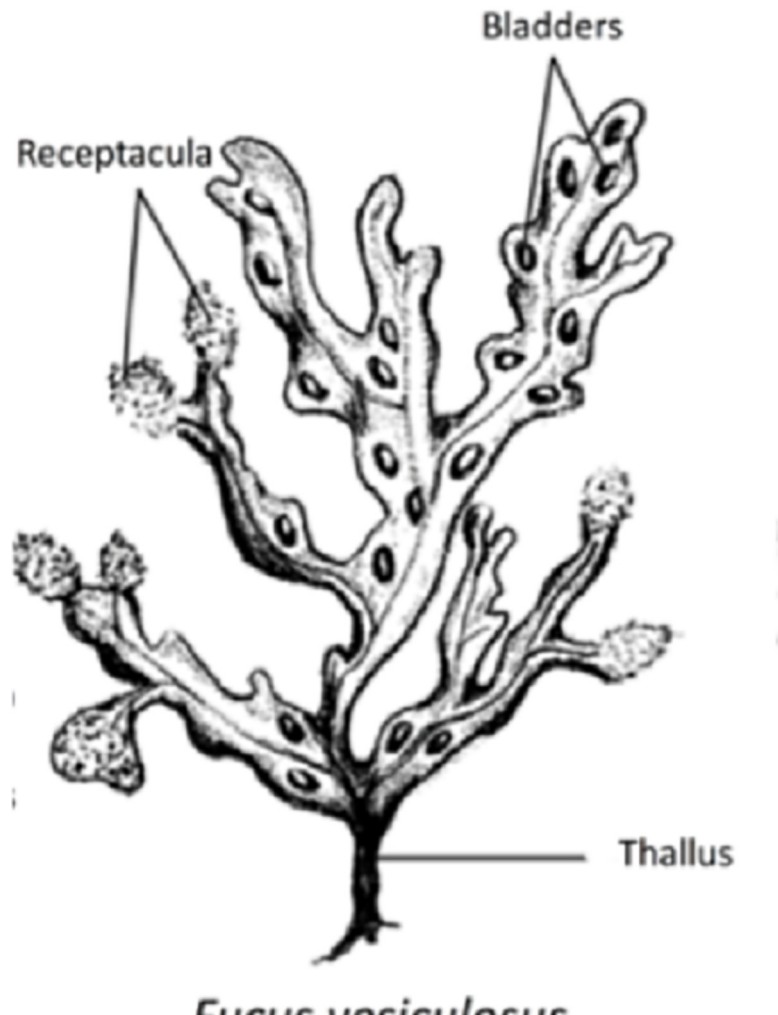

**Fig 2. Different algal structures of *Fucus vesiculosus*: Receptacula, floating bladders and thalli.** Figure modified from Guden et al. [50].

sieve, then preserved it in DESS (dimethyl sulphoxide disodium EDTA [57]) prior to counting. DESS not only preserves nematode morphology but also DNA, allowing later identification of cryptic nematode species using qPCR. After nematodes had thus been removed from their surface, the algal structures were incubated on sloppy marine agar medium (0.7% agar, 10/1 B/N, salinity of 15, see above) [53] at 20˚C in the dark, allowing nematodes from inside the receptacula and floating bladders to move out and colonize the agar. The nematodes on the agar plates were counted after 4 days of incubation under a Leitz Dialux stereomicroscope (14x – 60x magnification). Additional checkups of the agar plates were made after 11, 18 and 25 days in those cases where we did not see nematodes during previous observations.

To ascertain the presence of *L. marina*, we randomly picked up 10 individuals (if present) from each plate with nematodes for identification to species level using a quantitative real-time PCR of the ribosomal ITS region; this PCR was performed with four different primers, each of which matches exclusively with one of the four cryptic species of *L. marina* known from our study area [58]. As such, we identified 100 nematodes from inside receptacula, and 15 nematodes from inside floating bladders.

**Data analysis.** Nematode densities per $cm^2$ on the outer surface and on the inside of the receptacula and floating bladders (only data after 4 days of incubation on agar plates) were compared using a linear mixed model, implemented in the R software program (version 4.0.3, R Core Team 2020) with packages nlme version 3.1 [59] and lme4 version 1.1 [60], with algal structure and inside vs. outside as fixed factors. Replicates were included as a random factor in the model to account for the fact that some treatments were linked to each other (investigated on the same piece of alga, i.e. nematode abundance outside vs. inside). The model was evaluated by checking random scatter of the residuals against the fitted values. To achieve this, the density data had to be log transformed.

## Results

### Can marine nematodes disperse through air?

After one week of incubation in the field, no nematodes were observed in the boxes that were closed with the 200-µm gauze, while mostly dead nematodes were observed in the seawater from all the open boxes (Fig 3). No living nematodes were recorded in the agar plates whereupon we had inoculated the algae after the field experiment.

**Air dispersal versus dispersal through the water column.** In the second experiment, a significantly larger number of nematodes colonized the 'air only' boxes (average = 9772.6 ± 1255.1 SE) compared to the 'air + seawater' boxes (average = 210.9 ± 46.3 SE) (df = 1; F = 115.91; P = 1.86 x $10^{-8}$), which could become colonized through a tidal inflow of seawater as well (Fig 4).

We identified *Litoditis* sp. in both treatments, indicating that this nematode can disperse through the air. In fact, *L. marina* accounted for at least 26% of the total nematodes in the 'air only' boxes, and was also present in the 'air + seawater' boxes.

### Can macroalgae serve as a shelter for nematodes while rafting?

Nematodes were present inside floating bladders and receptacula, where they are likely more sheltered against water currents, as well as on the outside of the respective structures (Fig 5). There were significantly more nematodes on/in receptacula than on/in floating bladders

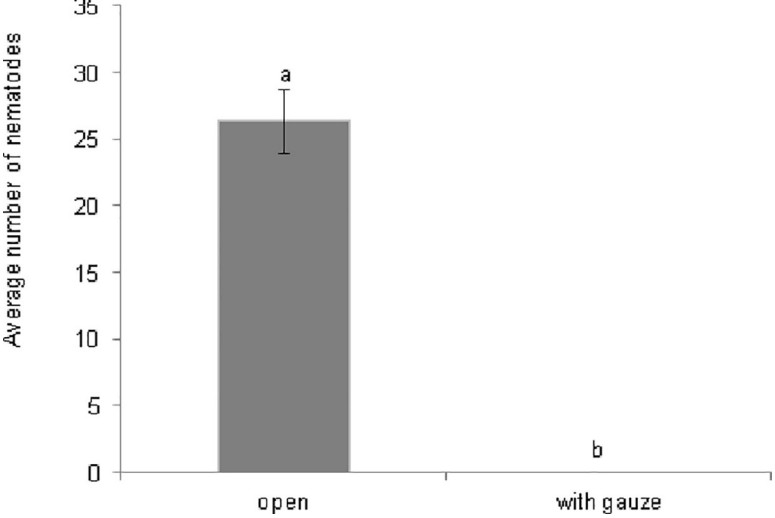

**Fig 3. Average numbers of nematodes (mean ± SE) in the seawater (ca. 100 ml) in the open or gauzed (200 µm) boxes.** Letters indicate significant differences between treatments.

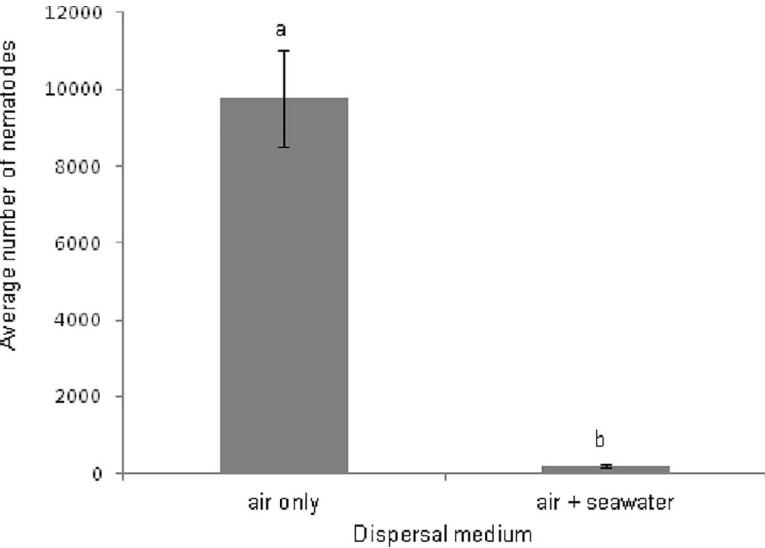

**Fig 4. Average numbers of nematodes (mean ± SE) in differently positioned boxes of the field experiment, which could be reached only through the air ('air only' boxes) or through the air and the water column ('air + seawater' boxes).** Letters indicate significant differences between treatments.

(df = 1; F = 24.90; P = 0.0001), but this difference could largely be attributed to a higher abundance of nematodes on the outside of the receptacula compared to the outside of the floating bladders (P = 0.001). None of the other pairwise comparisons for the interaction factor structure x inside/outside (df = 1; F = 4.45; P = 0.047) proved statistically significant (S1 Table).

42% of the nematodes associated with the receptacula could be identified as *L. marina* cryptic species Pm I using qPCR, but none of the few nematodes (15 specimens identified) associated with the floating bladders belonged to the *L. marina* species complex.

## Discussion

Although the overall dispersal rates of nematodes can be high, mainly as a consequence of bedload transport [7–10], per capita dispersal is low, particularly when compared to meiofauna

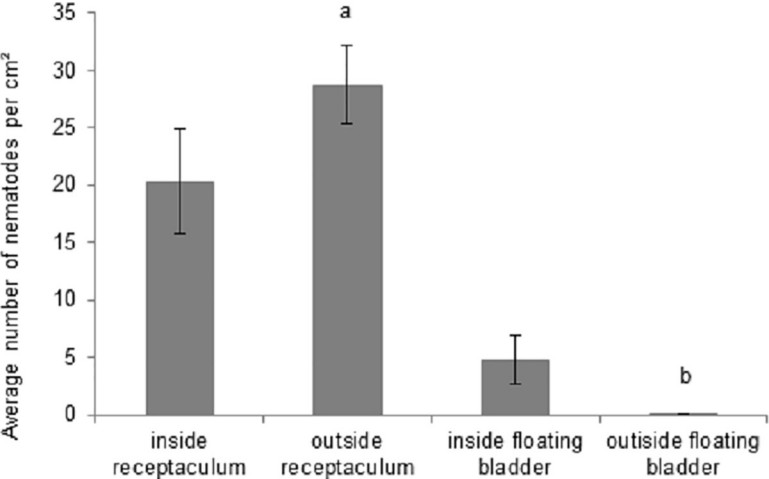

**Fig 5. Nematode abundance per cm$^2$ (mean ± SE) on the inside and outside of the receptacula and floating bladders of *Fucus vesiculosus*.** Different letters above the bars indicate significantly different means.

with active emergence behavior, such as copepods [10]. It is therefore important to investigate other forms of dispersal for nematodes, including airborne transport and rafting.

## Aerial dispersal of nematodes

Our first two experiments demonstrate that algal patches in an estuarine salt marsh can become colonized by nematodes that disperse through air. Moreover, our first experiment suggests that nematodes did not merely disperse through the action of wind, but were associated with an animal vector, since no nematodes were found in boxes that were covered with gauze that excluded such vectors. Transport of terrestrial and freshwater nematodes by birds [61, 62] and insects [38, 63, 64] has been repeatedly demonstrated, but we know of no other reports of aerial dispersal of any marine nematode. We found insects and spiders as well as bird feces on and inside the boxes, indicating that these may all have a potential role as vectors. Spiders have not hitherto been reported as vectors of any free-living nematodes, but this option cannot be excluded. Hitchhiking through air on invertebrate vectors provides an efficient dispersal mechanism beyond random dispersal for nematodes associated with deposits of algal wrack, because macroalgal patches are sparsely distributed and ephemeral habitats. Vector-borne aerial dispersal in *L. marina* is plausible, since many forms of nematode-arthropod relationships have been demonstrated in terrestrial and coastal members of the Rhabditida: associations of Rhabditida with arthropods include parasitism on insects [63] and vector-mediated disperal by crabs [33], Amphipoda [65, pers obs on *L. marina* sp.] and Isopoda [66]).

The fact that only dead nematodes were obtained from the boxes in the first experiment indicates that environmental conditions in the boxes rapidly turned unfavourable, possibly through a pronounced increase in the salinity of the ASW following evaporation. It is possible that environmental conditions also differed between boxes covered with gauze and open boxes; however, even if this would have turned the conditions in the gauzed boxes less favourable to nematodes, we should have found dead specimens as in the open boxes, but this was not the case.

Our second experiment demonstrated that nematode colonization of algal deposits through air can be as effective as through the water column, since much greater nematode abundances were encountered inside boxes which were out of reach of seawater than in boxes which were regularly flooded. Comparing these numbers nevertheless requires caution, since the high nematode abundances inside 'air only' boxes are undoubtedly a result of the successful reproduction of dispersers on these algae; indeed, *L. marina* may produce several hundred progeny per gravid female [35, 36]. Hence, it is impossible to estimate the number of nematodes that dispersed through air. Still, it is remarkable that dispersers did not establish similarly high population abundances in the 'air + seawater' boxes, because these could also be reached by vectors. One potential explanation is that flooding events may have washed away a large proportion of recently settled nematodes in these 'air + seawater' boxes [21, 46]. Alternatively, a larger volume of water retained in these boxes may have offered a less favorable environment for reproduction of colonizing nematodes (Moens, pers obs). Finally, although the horizontal distance between the 'air only' and the 'air + seawater' boxes was small (< 20 m), and the prevailing wind direction went from the 'air only' to the 'air + seawater' boxes, we cannot completely rule out the possibility that vectors visited the latter boxes less frequently, for instance because they were placed amidst a less diverse vegetation and/or retained more water.

## Macroalgal structures offer shelter for nematodes

Nematodes were observed inside both receptacula and floating bladders of *F. vesiculosus*; even after 18 days of incubation on agar, we still found nematodes inside these structures. As such,

these algal structures may offer protection against water currents and both prevent nematodes from being washed off their algal substrata and facilitate their dispersal over long distances through rafting, as confirmed by the first author upon collecting algae that were adrift in the Oosterschelde estuary (The Netherlands) in autumn 2013 (see below). We found more nematodes associated with the slimy and bacteria-rich receptacula [50, 67, 68]–potentially excellent feeding sites for bacterivorous nematodes–than with the floating bladders, but this difference was largely due to the nematode abundances on the outside of these structures. The significantly higher abundances of nematodes on the outside of receptacula than on floating bladders probably reflects the coarser, more irregular surface of the former, which may facilitate crawling and even attachment of nematodes to the surfaces of receptacula [50].

The hypothesis that nematodes may raft inside floating bladders was confirmed by the first author of this study upon collection of some floating algae (*Ascophyllum nodosum*) in the Oosterschelde estuary (The Netherlands) in September 2013. Three of four bladders contained nematodes, albeit very few (an average of 3 nematodes per bladder). In addition to *L. marina*, another marine nematode species (*Adoncholaimus* sp.), belonging to a completely different clade and not a typical associate of macroalgae, was also found in this anecdotal sampling [49], indicating that rafting inside macroalgae may provide a means of dispersal to more marine nematode species than only *L. marina* and/or other typical inhabitants of macroalgae.

## Conclusion

Our experiments demonstrate that at least some nematodes which inhabit marine intertidal habitats, like the morphospecies *L. marina*, can rapidly colonize suitable habitat patches such as freshly deposited macroalgal wrack through the air, most probably through hitchhiking on insect vectors. Further dedicated research will have to reveal which specific associations exist, whether other vectors that actively move in between suitable habitat spots (such as crabs) could also serve the same role, transporting nematodes from and to suitable habitat patches, and what is the quantitative importance of such vector-mediated dispersal relative to dispersal through bedload transport. Further research will also have to elucidate whether this mechanism is restricted to marine Rhabditidae or whether the phenomenon is more broadly used by nematodes from intertidal environments.

Our results also demonstrate that marine nematodes can live inside macroalgal structures such as receptacula and floating bladders, which may protect them from being washed off their algal substratum and thus allow these nematodes to raft over large distances with drifting macroalgae. As such, the results of the present study can be important for our understanding of both large-scale geographic distribution patterns and of the small-scale colonization dynamics of habitat patches by marine nematodes.

## Supporting information

**S1 Table. Statistical table of the main effects of the linear mixed model ANOVA with algal structure and inside vs. outside as fixed factors.** Replicates were included as a random factor. Pairwise comparisons of the interaction effect are also given. Significant effects are plotted in bold.
(DOCX)

## Acknowledgments

We acknowledge N. Viane, B. Beuselinck, A. Rigaux, M. Taheri and W. Stock for assistance in the field (set up and sampling of the experiments), L. Monteiro and X. Wu for help with

nematode identification, and A. Rigaux for help with the processing of samples in the laboratory, including the qPCR analyses.

## Author Contributions

**Conceptualization:** Bartelijntje Buys, Sofie Derycke, Tom Moens.

**Formal analysis:** Bartelijntje Buys, Sofie Derycke, Nele De Meester.

**Funding acquisition:** Sofie Derycke, Tom Moens.

**Investigation:** Bartelijntje Buys, Sofie Derycke.

**Methodology:** Bartelijntje Buys, Sofie Derycke, Nele De Meester, Tom Moens.

**Resources:** Tom Moens.

**Supervision:** Sofie Derycke, Tom Moens.

**Validation:** Bartelijntje Buys, Sofie Derycke, Nele De Meester, Tom Moens.

**Writing – original draft:** Bartelijntje Buys.

**Writing – review & editing:** Bartelijntje Buys, Sofie Derycke, Nele De Meester, Tom Moens.

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
