## [Decision Letter · Decision Letter 0]

10 Jul 2020

PONE-D-20-18326

Colonization of macroalgal deposits by estuarine nematodes through air and potential for rafting inside algal structures

PLOS ONE

Dear Dr. Moens,

Thank you for submitting your manuscript to PLOS ONE. After careful consideration, we feel that it has merit but does not fully meet PLOS ONE’s publication criteria as it currently stands. Therefore, we invite you to submit a revised version of the manuscript that addresses the points raised during the review process.

Please address all comments raised by the reviewers, particularly with respect to coverage of the appropriate literature.

We look forward to receiving your revised manuscript.

Kind regards,

Maura (Gee) Geraldine Chapman, PhD DSc

Academic Editor

PLOS ONE

Journal Requirements:

Reviewers' comments:

Reviewer's Responses to Questions

**Comments to the Author**

1. Is the manuscript technically sound, and do the data support the conclusions?

Reviewer #1: Yes

Reviewer #2: Yes

Reviewer #3: Partly

Reviewer #4: Yes

2. Has the statistical analysis been performed appropriately and rigorously? 

Reviewer #1: Yes

Reviewer #2: Yes

Reviewer #3: Yes

Reviewer #4: Yes

3. Have the authors made all data underlying the findings in their manuscript fully available?

Reviewer #1: Yes

Reviewer #2: No

Reviewer #3: Yes

Reviewer #4: No

4. Is the manuscript presented in an intelligible fashion and written in standard English?

Reviewer #1: Yes

Reviewer #2: Yes

Reviewer #3: Yes

Reviewer #4: Yes

5. Review Comments to the Author

Reviewer #1: This paper presents results of three experiments designed to evaluate the extent to which marine nematodes may be dispersed by aerial means or within the water column and whether macroalgae can act as a shelter for nematodes thus enabling rafting.

In the first experiment aerial dispersal was evaluated using three treatments in which defaunated and rehydrated algae were placed in boxes fixed to poles elevating them above the level of high tide and either; totally exposed to the air, partially enclosed to exclude flying vectors such as insects or birds or totally enclosed as control. After one week the number of nematodes on the algae and/or trapped water in each treatment was determined.

In the second experiment a similar arrangement was employed but in this case four boxes were exposed to air only while four were placed such that they would be inundated (or splashed) during each high tide over a period of two weeks.

The third experiment assessed the density of nematodes on the thalli of F. vesiculosus (specifically the bladders and receptacula) collected in the field.

The experimental protocols used were appropriate as were the analyses and presentation of the results. This was particularly evident in the air dispersal vs. water column experiment where the statistical analysis was virtually redundant. The results clearly show that aerial dispersal can be an important pathway for dispersion in marine nematodes. While the authors also showed that algae can act as rafts for nematode dispersal, I was not really convinced that a significant difference (P = 0.047) in abundance existed between bladders and receptacula.

I think this paper makes a useful contribution to our understanding of dispersal in marine nematodes and should be published.

Two minor issues: 1. There appears to be an error in one of the dimensions given in Fig 1 where interspace left is shown as 1210cm. 2. I think the authors should use consistent labelling of treatments in Fig 4. instead of "high" or "low" use aerial or aquatic as in the text.

Reviewer #2: Introduction

I would suggest to incorporate the new study by Ingels et al, 2020 on the dispersal of nematodes on Loggerhead sea turtles

Ingels, J.; Valdes, Y.; Pontes, L.P.; Silva, A.C.; Neres, P.F.; Corrêa, G.V.V.; Silver-Gorges, I.; Fuentes, M.M.; Gillis, A.; Hooper, L.; Ware, M.; O’Reilly, C.; Bergman, Q.; Danyuk, J.; Sanchez Zarate, S.; Acevedo Natale, L.I.; dos Santos, G.A.P. Meiofauna Life on Loggerhead Sea Turtles-Diversely Structured Abundance and Biodiversity Hotspots That Challenge the Meiofauna Paradox. Diversity 2020, 12(5), 203; https://doi.org/10.3390/d12050203.

Material and methods

You do not mention in any of the three experiments for how long it was conducted. Please add this information.

How do you know that the nematodes recovered after 7 and 14 days of experiment are not coming from a new generation, considering that their life cycle last for four days?

Results

Please provide your statistical results as a supplement table.

Line 278: “regularly observed”. How regular? I found that some parts of the manuscript lack clarity in the information it brings. Please try to include more accuracy in your statements.

Line 291: why would a small body size hamper dispersal? Many studies on meiofauna and macrofauna showed that they can disperse across large geographical distances so I do not understand this statement.

Discussion

Line 349: I agree that dispersal through vectors can occur in the terrestrial environment, but in the context of your study, which candidates can you suggest that thrive between the marine and “terrestrial” environment that can explain the high densities observed in the treatments located above the maximum tidal level?

Reviewer #3: General comments:

The authors conducted 2 field experiments to investigate estuarine nematode dispersal through the air by wind and animal vectors into boxes containing defaunated macroalgal wrack. They also examined the insides and outsides of algal floating bladders and receptacula for the presence of nematodes to determine the potential for dispersal by algal rafting. Results demonstrated that (1) open boxes contained nematodes, but boxes covered with gauze (to exclude animal vectors) did not; (2) boxes located above the high tide line contained more nematodes than did the boxes located lower in the saltmarsh; and (3) more nematodes were found inside and outside receptacula than inside and outside floating bladders. The results are somewhat compromised by possible research design problems. These are not fatal problems, but they need to be addressed more fully by the authors.

The authors discuss the results in relation to the so-called “meiofauna paradox,” i.e., the fact that meiofauna have poor dispersal abilities yet can have extensive geographic ranges. They claim that long-distance dispersal by aerial vectors and rafting is necessary to explain the apparent paradox. I agree that these forms of long-distance dispersal occur and may be important. However, bedload transport has been shown repeatedly to be an important dispersal mechanism for meiofauna, including nematodes. They have high rates of dispersal in bedload. These short-distance but frequent (daily) dispersal events can add up to long distances over time. The results of the current investigation need to be incorporated into this existing body of knowledge about nematode dispersal, which the authors only briefly mentioned, ignoring a number of very relevant publications. Doing so will strengthen the rationale for their research. For example, although the literature shows that absolute dispersal rates for nematodes are high, their per capita rate is lower than the rate for meiofaunal copepods. Thus, it is important to investigate other forms of dispersal for nematodes, including airborne and rafting modes of nematode dispersal.

The manuscript is clearly written, well organized, and easy to read. Please see specific comments below.

Specific comments:

Ln 44 – “Marine and estuarine organisms mainly disperse through active swimming (e.g. fish) or through drifting planktonic larvae (e.g. many macrobenthos species).”

The authors are correct about rapid dispersal over long distances. However, many benthic ecologists no longer agree with this statement when considering rapid dispersal over short distances or slow dispersal over long distances. See below.

Ln 46 – “Still, even species with limited swimming capacities or without pelagic larvae can disperse over variable distances through drifting (floating passively, e.g. through larva propagules), rafting (attached to a floating object), hitchhiking (attached to a mobile object or animal), hopping (going from one suitable habitat/substratum to another) or creeping (over substrates/habitats which are similar over a considerable distance) [7].”

Yes, all of these mechanisms can be important. However, the authors neglected to mention dispersal of juvenile and adult animals via hydrodynamic action, i.e., in bedload transport along the bottom. In the coastal zone, this form of dispersal occurs during every tidal cycle every day of the year, as well as by daily wind-generated water movement. Frequent, short distances can add up to long cumulative distances. Bedload transport is probably the most important dispersal mechanism for small benthic organisms (like nematodes) that lack a pelagic larval stage. Even macrofauna species with larval dispersal exhibit significant postlarval dispersal via bedload transport. For quantitative examples of nematode transport in bedload (daily transport rates and turnover times) as well as recent summaries of the literature on meiofauna dispersal via bedload transport, see Commito and Tita 2002 JEMBE, Commito et al. 2018 JEMBE, Commito et al. 2019 Diversity.

Ln 51- 63 – This sections seems to ignore much of the recent literature on meiofauna dispersal via bedload transport.

Ln 62 – “Recent studies suggest that effective meiofaunal dispersal is nevertheless restricted in space, even at scales of (tens of) kilometers [18-22]. This seems at odds with their often cosmopolitan distributions [8,13]. This so-called meiofauna paradox can be explained by occasional long distance dispersal events [23,24], leading to a wide geographic distribution of species.”

I do not see this as a meiofauna paradox. Frequent, short-distance dispersal events can add up to long distances over ecological, evolutionary, and geological time. Many of the world’s most cosmopolitan non-meiofauna benthic species have no free-swimming larval stage, e.g., the bivalve Gemma gemma, the oligochaete Tubificoides benedeni, and many opportunistic polychaetes. Like nematodes, these organisms are small and as a result cannot produce enough pelagic larvae for sufficient numbers to survive in the water column before settling. From an evolutionary perspective, dispersal as juveniles and adults is an effective strategy for small species, both meiofauna and non-meiofauna alike.

A good example is the small bivalve Gemma gemma. It broods its young and has no pelagic larval stage. Juvenile and adult Gemma are transported passively by bedload transport. Despite its limited dispersal ability, this coastal species is abundant for thousands of kilometers of coastline from Canada to the Gulf of Mexico, with little genetic variation between distant locations, indicating genetic mixing (Commito et al. 1995 Ecological Monographs, Casu et al. 2005 JEMBE).

Ecological time is relatively short, but evolutionary and geological time frames are long. It is possible that birds and other vectors may sometimes carry Gemma (and nematodes) long distances. However, daily short-distance bedload dispersal may carry organisms just as far as infrequent long-distance dispersal events. Nematodes are much smaller than Gemma, so they are probably even more easily eroded and dispersed by bedload transport.

Ln 95 – “nematodes hide inside algal structures”

Hide from what? Here and elsewhere in the manuscript, perhaps it is more accurate to say that nematodes “live” inside algal structures. To say they “hide” there seems to imply that they are avoiding detection from predators, which may be true, but we just do not know.

Ln 107 – “both experiments”

“Both” means 2 experiments. But there were 3 experiments.

Ln 120 – “Can marine nematodes disperse through air?”

In the Methods section, the authors present this part of their research as a question, as they do for the third (last) part. But not the second part. Please change so that they are all three parts are presented in the same way. And I suggest that the authors be consistent and use the same way for the Results and Discussion, too.

Ln 124-126 – “The algae collected at Paulina”

(1) What was the species composition of the wrack? Did the authors mix all the collected wrack before assigning it to experimental boxes?

(2) Can the authors provide evidence here that the defaunation protocol did indeed kill all the nematodes? I suggest that they move the Ln 171-176 information to this position in the manuscript.

In Ln 171-176, the authors said that they found no nematodes in the agar when testing pre-experiment defaunation of the algae used in their experiment boxes. But that is not a good test of pre-experiment defaunation. They did not find nematodes in the agar from any of their experimental treatments at the end of the experiment, indicating that looking in the agar was not a good way to determine if the nematodes had been eliminated beforehand.

Ln 127 – Remove the word “squared”

Ln 128 – Change “pinched” to “pitched.” Also, were the boxes watertight, or were there gaps between the box sides and floor that could have allowed water (and nematodes) to escape? This is an important question because the only nematodes that were counted in the data analysis were individuals found in the box water.

Ln 137 – “leaving free space between box and roof for visiting animals”

(1) The term “free space” is confusing here. If I am reading this correctly, there was no free space because the gauze was intended to prevent all animals (insects, spiders, mammals, birds) from entering the box.

(2) The gauze almost certainly created artifact effects by altering light, temperature, and moisture. These had direct impacts on the nematodes as well as indirect effects on nematodes due to differences in the algae. The authors need to address this concern.

Ln 140 – “Finally, (3) closed boxes were completely closed by a wooden lid directly on top of the box”

This closed box treatment is mentioned here, but no closed box data are presented in the Results section. I suggest that the closed box treatment be deleted from the manuscript.

Ln 144 – “We placed 4 open, 4 gauzed and 4 closed boxes in a randomized design in the field for one week”

(1) Please describe the randomization procedure.

(2) A randomized design is fine here, although a randomized block design may have been more powerful, especially if there were position effects in the field.

(3) Here and below, the experiments were run for one week. That seems like a reasonable length of time. However, nematodes have a life cycle shorter than one week, so the number of nematodes in a box was the net result of dispersal into the box, production of new young, and mortality. At the end of the experiment, the authors were not actually counting the number of nematodes that had dispersed into the boxes. The differences between treatments may have been due to favorable or unfavorable environmental conditions within the different kinds of boxes. For example, the boxes with gauze may have held in more moisture, which may have affected the number of nematodes. Can the authors address these points?

Ln 181 – “the average amounts of nematodes”

Here and elsewhere in the text and the vertical axes of the figures, the authors should use “number” rather than “amounts” because nematodes are countable objects, not measured as a non-countable mass or volume. The figure legends use correct terminology.

Ln 190 – Can the authors comment on how the season of each experiment (September vs April) may have affected the results?

Ln 193 – “Four open boxes with rehydrated defaunated algae were mounted onto 3-m high sticks and placed ca. 1.75 m above the maximal tide level, while four others were positioned closer to the seaward edge of the marsh at ca. 0.27 m below the average tidal level.”

(1) Relative to the locations of the 2 sets of boxes, where was most of the naturally occurring macroalgal wrack located?

(2) Note that classic experiments on airborne dispersal of small aquatic organisms has been studied by placing sampling units at different heights on the same pole (e.g., Maguire 1963 Ecological Monographs, Maguire 1971 Annual Review of Ecology and Systematics). Can the authors explain why they decided to introduce a spatial variable into their experiment by placing the “air” treatment boxes above the high tide line and the “air + water” treatment boxes lower in the tidal zone? Would it have been possible to place boxes for both treatments on the same sticks, with the “air” treatment boxes higher on the stick than the “air + water” treatment boxes? That would have allowed a nice paired t-test design and avoided possible spatial difference complications briefly mentioned by the authors.

(3) The experimental design is presented as “air” vs “air + water” effects on nematode dispersal. It could just as easily be described as an experiment measuring “high marsh” vs “low marsh” effects on nematode dispersal or “diverse vegetation” vs “monospecific Spartina vegetation” effects on nematode dispersal. In fact, Fig. 4 labels the 2 treatments not as “air” vs “air + water,” but as “high” and “low”! The authors need to address this problem in more detail.

(4) Can the authors provide data on insect abundance and wind direction? If insects and wind are vectors, then they would strongly affect nematode dispersal into boxes at the 2 different locations.

Ln 204 – “We sampled all boxes after 14 days of exposure in the field.”

See second comment above for Ln 144 re: length of exposure period compared to length of nematode life cycle, as well as temperature and moisture differences between the treatment boxes.

Ln 228 – “We randomly collected 17 complete thalli of the brown alga Fucus vesiculosus at the edge of the Paulina salt marsh”

(1) Please describe the randomization procedure.

(2) Which edge of the marsh, upper or lower?

(3) Were these samples taken from live algae or from wrack?

Ln 231-236 – “Each structure was then thoroughly rinsed with tap water to remove the nematodes present on the outside of the structures, before incubating it on sloppy marine agar medium (0.7% agar, 10/1 B/N, salinity of 15, see above) [47] at 20 °C in the dark. The water containing nematodes from the outer surface of each of these structures was collected separately over a 38-μm sieve and preserved in DESS [50] prior to counting.”

(1) Confusing. I do not quite understand what the authors did here. I see no mention of “inside” the structures.

(2) Please define “DESS.”

Ln 281 – “Nematodes were present inside floating bladders and receptacula, although statistically not more than on the outside of the respective structures (Fig 5).”

(1) Is there any reason to think that the insides would have more than the outsides? That was not one of the research questions the authors set out to answer.

(2) See misspelling on horizontal axis of Fig. 5 “outiside” should be “outside”)

Ln 291 – “The active dispersal capacity of marine nematodes is limited because of their small body size, limited swimming ability and lack of larval stages that can disperse (reviewed in [13]). Hence, dispersal over both short and long distances is generally believed to be largely passive [10,13] and to result from transport through the water column [11]. Alternative dispersal mechanisms in marine nematodes have received only limited attention [16,17].”

Yes, nematode dispersal is largely passive. But that does not mean that such dispersal is rare or unimportant. Please see comments above for Ln 46. Bedload transport (rather than water column transport per se) has been shown repeatedly to be an important dispersal mechanism for marine nematodes, and it is strongly controlled by wind-generated hydrodynamic forces. These dispersal rates have been shown to vary according to nematode feeding type. Some feeding types live closer to the bed surface and are more easily eroded and transported. In fact, epigrowth-feeders have by far the highest absolute, relative, and bulk dispersal rates.

Overall, nematodes have very high absolute dispersal rates, as would be expected given their abundance in marine sediment. However, when normalized to a per capita basis, they do not disperse at rates as high as meiofauna with active emergence behavior, such as copepods. (See Discussion in Commito et al. 2019 Diversity for comparison of per capita bedload transport rates of nematodes vs copepods, with data from a variety of locations.) Thus, it is important to investigate other forms of dispersal for nematodes, including airborne and rafting modes of nematode dispersal.

Ln 325 – “Hitchhiking through air on invertebrate vectors provides a more direct dispersal mechanism for nematodes associated with deposits of algal wrack, and may have many benefits above random dispersal through water currents, because macroalgal patches are sparsely distributed and often ephemeral habitats, and because nematodes lack good active swimming capacities, whilst at least some of their candidate vectors can efficiently move between such patches.”

(1) For nematodes that can live in macroalgal patches and in sediments, the nematodes may move by bedload transport directly from sediment onto macroalgal patches. They may also “hop-scotch” from a macroalgal patch to the sediment and then from the sediment to another macroalgal patch. The authors need to address this point.

(2) Change “above random dispersal” to “beyond random dispersal” or “in addition to random dispersal”

(3) Water currents certainly have a random component due to turbulence. But water currents typically have strong directional component due to the ebb and flow of the tides as well as to wind direction. These water currents move meiofauna, including nematodes, in predictable (not random) directions (e.g., Fegley 1988 JEMBE, DePatra and Levin 1989 JEMBE, Commito et al. 2019 Diversity).

Ln 335 – Please define “entomopathogenicity” and “phoresy,” which I do not think most PLOS ONE readers will know.

Ln 339-355 – I am glad that the authors presented several possible explanations why the “air” boxes had higher numbers of nematodes than did the “air + water” boxes. The 2 treatments were established in 2 very different habitats as well as at 2 different distances from the marine source of nematodes. So the experimental results are somewhat compromised by the experimental design, as mentioned above in my comments for Ln 193. However, the results do point the way to future research.

Ln 373 – “Nematodes may search for protection on and inside the latter structures”

Perhaps nematodes exhibit this type of searching behavior, but do the authors believe it is likely that nematodes actively search for such protection? Is there any evidence for it in the literature?

Ln 386 – “Further dedicated research will have to reveal which specific associations exist, and whether other, more sediment-bound vectors that actively move in between suitable habitat spots (such as crabs) could also serve the same role, transporting nematodes from and to suitable habitat patches.”

See comments above for Ln 46, 291, and 325. Bedload transport is an example of a “more sediment-bound vector.”

Ln 396 – “This may explain existing evidence for “long-distance dispersal events” in Litoditis marina and perhaps other marine nematodes, and may therefore provide one answer to the meiofauna paradox: meiofauna lack larval dispersal stages or active larger-scale dispersal ability, yet quite many species have very extensive geographic distributions. As such, the results of the present study can be important for our understanding of both large-scale geographic distribution patterns and of the small-scale colonization dynamics of habitat patches by marine nematodes.”

I certainly agree that drifting macroalgae can be a long-distance dispersal mechanism for nematodes. However, as mentioned above in my comments for Ln 62, I do not believe that a meiofaunal paradox exists. A species does not need long-distance dispersal events in order to have an extensive geographic distribution. Daily short-distance bedload dispersal may carry organisms just as far as infrequent long-distance dispersal events.

Reviewer #4: Review of Colonization of macroalgal deposits by estuarine nematodes through air and potential for rafting inside algal structures. PONE-D-20-18326.

This is an interesting and novel piece of fundamental ecological research and I recommend it for publication in PLOS One after minor revisions. Whilst there are no major flaws that I have identified (although some confounding in experiment 2: see comments below), the text is a little unclear in several places and requires some clarification to enhance the message. The discussion is also rather repetitious and could be shortened and tightened.

Minor comments:

Abstract: sentence from 30-33 is rather long-winded and it makes it difficult to read. I suggest to shorten this perhaps start “We also demonstrate for the first time….” And remove “and even quite efficiently”.

Line 60: what is meant by “subsequent phases”? do you mean recruitment? Please specify.

Line 70 onwards: after first mention, Litoditis marina can be referred to as L. marina.

Line 94: grammatically the “both” should come occur after “nematode dispersal” as in “we investigated nematode dispersal both through air…”

Line 96 to 103 seems out of place in an introduction and would perhaps be better in the methods since it is describing what was done.

Please add a hypothesis or hypotheses to the end of the introduction.

Line 123 and elsewhere: by “water” is “seawater” meant or rain? Please be specific.

Line 137: the mesh is 200um so which visiting animals is this allowing entrance for? I thought that the mesh was the allow wind dispersal but prevent vector dispersal on insects? This is a little unclear, please clarify.

Line 151: metal piles should be metal poles? Also in figure 1 and elsewhere.

Methods: Are the boxes water-tight? Are they lined with something to make them hold water?

Line 156: write the whole word for diameter.

Why were the nematodes dead in the first experiment and so few in abundance? Some further exploration of this in the discussion would be good. Was it due to the methods, did the boxes get too hot or dry out?

Line 194 and throughout: “Sticks” or “poles”? Please be consistent in some cases these are also called “piles”. Also what were these poles made out of?

Line 199: Some more detail is required on how much water these “low” boxes received. How long were they immersed for? Did the whole box become covered?

Line 200: treatments are called “air only” and “air + water” but are called “low” and “high” in figure 4. Please be consistent.

~line 216: how many nematodes or what % was accurately sampled?

Line 223: why log- transformed?

~Line 233: Bit unclear how the inside versus outside of the algal structures were sampled. Rinsed to remove outside, but then were the structures cut to release the nematodes from within? Is there a protocol to refer to that proves that this technique works? Is it possible that some nematodes hold on very strong and are still present on outside even after rinsing? Please provide some extra evidence here.

Line 241: the “dedicated qPCR protocol” needs some explanation here, it is ok to be brief, but simply referring to another paper for an entire method is too vague. Also how many samples were analysed in this way?

Line 251: “resp.?” please write in full.

Line 253: what programme was used for the analysis and which packages with references please.

Line 266: Suggest using “greater amount” instead of “higher amount” since this confuses with the height on the shore.

Line 284: “the few” is too vague. How many samples and what proportion?

Line 310: wind transport is discussed here but the mesh treatments in the first experiment had no nematodes thereby it was likely to be vector based transport? Bit confusing. I suggest moving the sentence in 315 upwards and moving or removing the sentence in 313. The flow of this discussion on this point is confusing and also there is repetition with the introduction so not all of this text is needed. Could remove from 310 to 315.

Line 328-331: Does this explain why there were so few nematodes in the water + air treatments? This section could be related more clearly to experiment 2.

Discussion: Why do the lower boxes have so few nematodes, could it be that the seawater is flushing them back out of the boxes as the water ebbs in and out? I think a sentence or two is needed in the discussion to address this explicitly. I see it occurs in lines 342 onwards, but think it needs to be earlier, so suggest to move this up to around line 328 and shorten the whole section.

Line 332-338: this seems repetitious too, please check for repetition with the intro and preceding discussion and try to reduce.

Line 340: greater instead of higher. This occurs a few times, grammatically it is clearer to use “greater”.

The treatments in the second experiment which aim to separate “water + air” with “air only” are confounded with height on the shore which also includes temperature differences as well as humidity and moisture and possibly fauna. One way to separate this could have been to wet the “high boxes” with fresh seawater each day for the same amount of time and volume that the “low boxes” received. Is there evidence that the insect or arthropod vectors visit the lower shore at all? The manuscript states that they “probably” do. Can this be supported by literature or at least acknowledged as an unknown? This confounding is partly acknowledged around lines 347 but more is required please.

Line 378: please remove the etc…..

Line 378: “robust” in what way?

Line 398-399: “quite many species” is vague, please replace.

All figures: use capital letters at start of axis titles.

Figure 2: do you have or require permission to use this image?

Figure 3: y-axis title is a bit too long, perhaps “Average abundance of nematodes”, the per box part can be in the figure legend.

6. PLOS authors have the option to publish the peer review history of their article (what does this mean?). If published, this will include your full peer review and any attached files.

Reviewer #1: **Yes: **Arthur Dye

Reviewer #2: No

Reviewer #3: **Yes: **John A. Commito

Reviewer #4: No

---

## [Author Response · Author response to Decision Letter 0]

24 Jan 2021

Manuscript PONE-D-20-18326

Response to Reviewers

Dear Dr. Maura (Gee) Geraldine Chapman, 

Thank you for giving us the opportunity to submit a revised draft of the manuscript “Colonization of macroalgal deposits by estuarine nematodes through air and potential for rafting inside algal structures” for publication in the journal PLoS One (document labeled “Manuscript”). We appreciate the time and effort that you and the reviewers dedicated to providing feedback on our manuscript and are grateful for the insightful comments on, and valuable improvements to our paper. We have incorporated most of the suggestions made by the reviewers. Those changes are highlighted in the document “Revised Manuscript with Track Changes”. Please see below for a point-by-point response to the reviewers’ comments and concerns. 

Reviewer #1:

 This paper presents results of three experiments designed to evaluate the extent to which marine nematodes may be dispersed by aerial means or within the water column and whether macroalgae can act as a shelter for nematodes thus enabling rafting.

In the first experiment aerial dispersal was evaluated using three treatments in which defaunated and rehydrated algae were placed in boxes fixed to poles elevating them above the level of high tide and either; totally exposed to the air, partially enclosed to exclude flying vectors such as insects or birds or totally enclosed as control. After one week the number of nematodes on the algae and/or trapped water in each treatment was determined.

In the second experiment a similar arrangement was employed but in this case four boxes were exposed to air only while four were placed such that they would be inundated (or splashed) during each high tide over a period of two weeks.

The third experiment assessed the density of nematodes on the thalli of F. vesiculosus (specifically the bladders and receptacula) collected in the field.

The experimental protocols used were appropriate as were the analyses and presentation of the results. This was particularly evident in the air dispersal vs. water column experiment where the statistical analysis was virtually redundant. The results clearly show that aerial dispersal can be an important pathway for dispersion in marine nematodes. While the authors also showed that algae can act as rafts for nematode dispersal, I was not really convinced that a significant difference (P = 0.047) in abundance existed between bladders and receptacula.

I think this paper makes a useful contribution to our understanding of dispersal in marine nematodes and should be published.

Response: Thank you for your interest. We appreciate the fact that you are stating the relevance of our work so clearly and comprehensively. We agree that the difference between nematode abundance on bladders versus receptacula is not unambiguous; while there is a highly significant difference in abundances of nematodes associated with receptacula vs bladders, this difference is mainly caused by the nematode abundances on the outside of the respective structures, whereas the differences for the inside of both structures are not statistically significant. We have corrected and clarified this point in results and discussion now.

Two minor issues: 1. There appears to be an error in one of the dimensions given in Fig 1 where interspace left is shown as 1210cm. 

2. I think the authors should use consistent labeling of treatments in Fig 4. instead of "high" or "low" use aerial or aquatic as in the text.

Response: 1. Our apologies. Figure 1 is now corrected. Interspace right side 12 cm, left side 10 cm. Pitched shelter roof.

2. We now clarified figure labels, legend and text. ‘High’ boxes are now labeled as ‘air only’ and ‘low’ boxes as ‘air + seawater’ boxes. 

Reviewer #2:

 Introduction

I would suggest to incorporate the new study by Ingels et al, 2020 on the dispersal of nematodes on Loggerhead sea turtles

Ingels, J.; Valdes, Y.; Pontes, L.P.; Silva, A.C.; Neres, P.F.; Corrêa, G.V.V.; Silver-Gorges, I.; Fuentes, M.M.; Gillis, A.; Hooper, L.; Ware, M.; O’Reilly, C.; Bergman, Q.; Danyuk, J.; Sanchez Zarate, S.; Acevedo Natale, L.I.; dos Santos, G.A.P. Meiofauna Life on Loggerhead Sea Turtles-Diversely Structured Abundance and Biodiversity Hotspots That Challenge the Meiofauna Paradox. Diversity 2020, 12(5), 203; https://doi.org/10.3390/d12050203.

Response: Thank you for this suggestion; very interesting paper indeed. This reference has now been added in the introduction as an example of long-distance dispersal events in marine nematodes.

Material and methods

You do not mention in any of the three experiments for how long it was conducted. Please add this information.

Response: We revised the manuscript, stating more clearly the duration of each experiment.

How do you know that the nematodes recovered after 7 and 14 days of experiment are not coming from a new generation, considering that their life cycle last for four days?

Response: We agree with the reviewer’s concern. It is very probable that the samples included individuals of in situ reproduction. This fact is now clearly specified in the discussion. 

We chose to sample at day 7 and 14 to increase the chance of nematode colonization in the field. In case air dispersal of marine nematodes would have been rather a rare/occasional phenomenon, we might not have detected the few nematodes in the relatively large decaying algae samples which we used to attract possible flying vectors. 

Results

Please provide your statistical results as a supplement table. 

Response: For the mixed model ANOVA, a supplementary table with the model statistics has now been added. For the t-tests, the values of the F statistic have been added in the text, but this was already the case in the original ms.

Line 278: “regularly observed”. How regular? I found that some parts of the manuscript lack clarity in the information it brings. Please try to include more accuracy in your statements.

Response: This remark is fully justified; we would have preferred to be able to provide more quantitative data for this specific observation. The observation described is based on first screenings (counts of nematodes) of all the samples prior to, and during the process of, making microscopic slides for nematode identification. Most unfortunately, the slides accidentally got lost in a lab clean-up before all or even most identifications were made. We therefore changed the wording 'regularly observed' to 'present', even though the limited portion of nematodes identified suggested a dominance of L. marina in the samples.

Line 291: why would a small body size hamper dispersal? Many studies on meiofauna and macrofauna showed that they can disperse across large geographical distances so I do not understand this statement.

Response: This part is now removed from the manuscript, also based on another reviewer’s comment. 

Our original statement was based on the often suggested idea that active dispersal capacities of nematodes relate to body size, in other words: larger nematodes would be more motile. But admittedly, small body sizes may favor passive or semi-passive (e.g., hitchhiking) dispersal over both local and larger scales.

Discussion

Line 349: I agree that dispersal through vectors can occur in the terrestrial environment, but in the context of your study, which candidates can you suggest that thrive between the marine and “terrestrial” environment that can explain the high densities observed in the treatments located above the maximum tidal level?

Response: We mainly think of insects and perhaps spiders living on decomposing material in intertidal habitats (see, e.g., Lanna Cheng, Marine insects, 1973, available at vliz.be). We found spiders, flies, bark lice, lacewings and bugs on the algae in the boxes at the day of sample collection. They were also present on decaying Spartina (cordgrass) ca 100 m away from our experimental plot. Birds are also possible vectors; on two occasions, we found bird droppings on the gauze and/or the roof of boxes.

Reviewer #3:

General comments:

The authors conducted 2 field experiments to investigate estuarine nematode dispersal through the air by wind and animal vectors into boxes containing defaunated macroalgal wrack. They also examined the insides and outsides of algal floating bladders and receptacula for the presence of nematodes to determine the potential for dispersal by algal rafting. Results demonstrated that (1) open boxes contained nematodes, but boxes covered with gauze (to exclude animal vectors) did not; (2) boxes located above the high tide line contained more nematodes than did the boxes located lower in the saltmarsh; and (3) more nematodes were found inside and outside receptacula than inside and outside floating bladders. The results are somewhat compromised by possible research design problems. These are not fatal problems, but they need to be addressed more fully by the authors.

The authors discuss the results in relation to the so-called “meiofauna paradox,” i.e., the fact that meiofauna have poor dispersal abilities yet can have extensive geographic ranges. They claim that long-distance dispersal by aerial vectors and rafting is necessary to explain the apparent paradox. I agree that these forms of long-distance dispersal occur and may be important. However, bedload transport has been shown repeatedly to be an important dispersal mechanism for meiofauna, including nematodes. They have high rates of dispersal in bedload. These short-distance but frequent (daily) dispersal events can add up to long distances over time. The results of the current investigation need to be incorporated into this existing body of knowledge about nematode dispersal, which the authors only briefly mentioned, ignoring a number of very relevant publications. Doing so will strengthen the rationale for their research. For example, although the literature shows that absolute dispersal rates for nematodes are high, their per capita rate is lower than the rate for meiofaunal copepods. Thus, it is important to investigate other forms of dispersal for nematodes, including airborne and rafting modes of nematode dispersal.

The manuscript is clearly written, well organized, and easy to read. Please see specific comments below.

Response: Thank you for your interest, insights and valuable comments on this manuscript. We took most of your concerns into account and incorporated the importance of bedload dispersal in our introduction, discussion and conclusion with references to four relevant papers. We also removed the term “meiofaunal paradox” from the ms.

Specific comments:

Ln 44 – “Marine and estuarine organisms mainly disperse through active swimming (e.g. fish) or through drifting planktonic larvae (e.g. many macrobenthos species).”

The authors are correct about rapid dispersal over long distances. However, many benthic ecologists no longer agree with this statement when considering rapid dispersal over short distances or slow dispersal over long distances. See below.

Ln 46 – “Still, even species with limited swimming capacities or without pelagic larvae can disperse over variable distances through drifting (floating passively, e.g. through larva propagules), rafting (attached to a floating object), hitchhiking (attached to a mobile object or animal), hopping (going from one suitable habitat/substratum to another) or creeping (over substrates/habitats which are similar over a considerable distance) [7].”

Yes, all of these mechanisms can be important. However, the authors neglected to mention dispersal of juvenile and adult animals via hydrodynamic action, i.e., in bedload transport along the bottom. In the coastal zone, this form of dispersal occurs during every tidal cycle every day of the year, as well as by daily wind-generated water movement. Frequent, short distances can add up to long cumulative distances. Bedload transport is probably the most important dispersal mechanism for small benthic organisms (like nematodes) that lack a pelagic larval stage. Even macrofauna species with larval dispersal exhibit significant postlarval dispersal via bedload transport. For quantitative examples of nematode transport in bedload (daily transport rates and turnover times) as well as recent summaries of the literature on meiofauna dispersal via bedload transport, see Commito and Tita 2002 JEMBE, Commito et al. 2018 JEMBE, Commito et al. 2019 Diversity.

Ln 51- 63 – This sections seems to ignore much of the recent literature on meiofauna dispersal via bedload transport.

Response to these first three specific comments: Thank you for these important comments; we added the importance of bedload transport both in the introduction, discussion and conclusion of our revised ms, and used the suggested literature references as well as two older ones by the same author. 

Ln 62 – “Recent studies suggest that effective meiofaunal dispersal is nevertheless restricted in space, even at scales of (tens of) kilometers [18-22]. This seems at odds with their often cosmopolitan distributions [8,13]. This so-called meiofauna paradox can be explained by occasional long distance dispersal events [23,24], leading to a wide geographic distribution of species.”

I do not see this as a meiofauna paradox. Frequent, short-distance dispersal events can add up to long distances over ecological, evolutionary, and geological time. Many of the world’s most cosmopolitan non-meiofauna benthic species have no free-swimming larval stage, e.g., the bivalve Gemma gemma, the oligochaete Tubificoides benedeni, and many opportunistic polychaetes. Like nematodes, these organisms are small and as a result cannot produce enough pelagic larvae for sufficient numbers to survive in the water column before settling. From an evolutionary perspective, dispersal as juveniles and adults is an effective strategy for small species, both meiofauna and non-meiofauna alike.

A good example is the small bivalve Gemma gemma. It broods its young and has no pelagic larval stage. Juvenile and adult Gemma are transported passively by bedload transport. Despite its limited dispersal ability, this coastal species is abundant for thousands of kilometers of coastline from Canada to the Gulf of Mexico, with little genetic variation between distant locations, indicating genetic mixing (Commito et al. 1995 Ecological Monographs, Casu et al. 2005 JEMBE).

Ecological time is relatively short, but evolutionary and geological time frames are long. It is possible that birds and other vectors may sometimes carry Gemma (and nematodes) long distances. However, daily short-distance bedload dispersal may carry organisms just as far as infrequent long-distance dispersal events. Nematodes are much smaller than Gemma, so they are probably even more easily eroded and dispersed by bedload transport.

Response: Thank you for these clear insights. We have removed the term “meiofauna paradox” and have reformulated the pertinent text to: 

“Recent studies suggest that effective meiofaunal dispersal nevertheless may be restricted in space, even at scales of (tens of) kilometers. This seems at odds with their often cosmopolitan distributions, but can be explained by occasional long-distance dispersal and the cumulative effect of relatively short-distance but frequent dispersal events through bedload transport, leading to a wide geographic distribution of species.” 

Ln 95 – “nematodes hide inside algal structures”

Hide from what? Here and elsewhere in the manuscript, perhaps it is more accurate to say that nematodes “live” inside algal structures. To say they “hide” there seems to imply that they are avoiding detection from predators, which may be true, but we just do not know.

Response: We agree with this statement, and have corrected the word ‘hide’ to ‘live inside’ or ‘reside in’ throughout the ms. 

Ln 107 – “both experiments”

“Both” means 2 experiments. But there were 3 experiments. 

Response: Correct. We have replaced ‘both’ by ‘all’ here.

Ln 120 – “Can marine nematodes disperse through air?”

In the Methods section, the authors present this part of their research as a question, as they do for the third (last) part. But not the second part. Please change so that they are all three parts are presented in the same way. And I suggest that the authors be consistent and use the same way for the Results and Discussion, too.

Response: OK, we have rephrased the title of the second experiment to: “Are decomposing macroalgae colonized by nematodes through air or through the water column?"

Ln 124-126 – “The algae collected at Paulina”

(1) What was the species composition of the wrack? Did the authors mix all the collected wrack before assigning it to experimental boxes?

(2) Can the authors provide evidence here that the defaunation protocol did indeed kill all the nematodes? I suggest that they move the Ln 171-176 information to this position in the manuscript.

Response: (1) By far the most abundant macroalgal species was Fucus vesiculosus, but we cannot exclude that some fragments of Fucus spiralis would have been included in the boxes. This info has been added to the revised ms.

 (2). We have moved the text fragment about the efficiency testing of the defaunation procedure forward as requested by this reviewer.

In Ln 171-176, the authors said that they found no nematodes in the agar when testing pre-experiment defaunation of the algae used in their experiment boxes. But that is not a good test of pre-experiment defaunation. They did not find nematodes in the agar from any of their experimental treatments at the end of the experiment, indicating that looking in the agar was not a good way to determine if the nematodes had been eliminated beforehand.

Response: We understand your concern but we respectfully disagree. We have been using incubation of macroalgae on agar slants as an effective means of isolating nematodes from the algae for many years now. We suspect that the reason why we could not isolate any nematodes from the algae which had been incubated in the field, was a deterioration of the ‘environmental’ conditions inside the experimental boxes. This view is supported by the fact that nematodes were able to colonize the algae in the boxes (as demonstrated by the presence of nematodes in the water inside the boxes), but that all nematodes found inside these boxes were dead.

Ln 127 – Remove the word “squared” 

Response: Agreed. The word “squared” has been removed.

Ln 128 – Change “pinched” to “pitched.” Also, were the boxes watertight, or were there gaps between the box sides and floor that could have allowed water (and nematodes) to escape? This is an important question because the only nematodes that were counted in the data analysis were individuals found in the box water.

Response: Pitched, check. Thank you for pointing this out. There were no gaps between the box sides and floor that could have allowed water (and nematodes) to ‘drain out’. We have therefore added the word ‘waterproof’ in the description of the boxes. 

Ln 137 – “leaving free space between box and roof for visiting animals”

(1) The term “free space” is confusing here. If I am reading this correctly, there was no free space because the gauze was intended to prevent all animals (insects, spiders, mammals, birds) from entering the box.

Response: We understand the confusion and have therefore clarified this as follows, among other things by removing the confusing wording “free space”:

“Gauzed boxes covered with a gauze with a mesh size of 200 µm, preventing entry of insects, spiders and other potential vectoring fauna, but allowing entry of nematodes transported by wind or dropping off from vectors landing on top of the gauze.” 

In other words, in case we would have found nematodes inside the gauzed boxes, this could have been because of (1) wind transport, (2) release of nematodes from insects perching on the gauze.

(2) The gauze almost certainly created artifact effects by altering light, temperature, and moisture. These had direct impacts on the nematodes as well as indirect effects on nematodes due to differences in the algae. The authors need to address this concern.

Response: The following information has been added to the discussion of the results of this first experiment: “The fact that only dead nematodes were obtained from the boxes in the first experiment indicates that environmental conditions in the boxes rapidly turned unfavourable, possibly through a pronounced increase in the salinity of the ASW following evaporation. It is possible that environmental conditions also differed between boxes covered with gauze and open boxes; however, even if this would have turned the conditions in the gauzed boxes less favourable to nematodes, we should have found dead specimens as in the open boxes, but this was not the case.”

Ln 140 – “Finally, (3) closed boxes were completely closed by a wooden lid directly on top of the box”

This closed box treatment is mentioned here, but no closed box data are presented in the Results section. I suggest that the closed box treatment be deleted from the manuscript.

Response: We followed the reviewer’s suggestion. The closed box treatment has been completely deleted from the manuscript.

Ln 144 – “We placed 4 open, 4 gauzed and 4 closed boxes in a randomized design in the field for one week”

(1) Please describe the randomization procedure.

Response: There was not really a randomization procedure; apologies for our inaccurate use of a word with a clear scientific meaning. We put them (gauzed/open/closed) haphazardly/randomly in the field after we had set up the poles. The pertinent text has been changed accordingly.

(2) A randomized design is fine here, although a randomized block design may have been more powerful, especially if there were position effects in the field.

Response: See response to the previous comment. The placement of the boxes did not follow a randomized design, but happened haphazardly within a square section of marsh measuring 14 x 6 m. 

(3) Here and below, the experiments were run for one week. That seems like a reasonable length of time. However, nematodes have a life cycle shorter than one week, so the number of nematodes in a box was the net result of dispersal into the box, production of new young, and mortality. At the end of the experiment, the authors were not actually counting the number of nematodes that had dispersed into the boxes. The differences between treatments may have been due to favorable or unfavorable environmental conditions within the different kinds of boxes. For example, the boxes with gauze may have held in more moisture, which may have affected the number of nematodes. Can the authors address these points?

Response: This is absolutely correct. Our main goal was to know whether or not marine nematodes may disperse through the air, and our experimental setup allowed to test that. However, a reliable quantification of this dispersal per se was not possible, since rhabditid nematodes have generation times that can be shorter than the field incubations (depending on, among other things, temperature). As a consequence, very high numbers of nematodes inside the “air only” boxes, for instance, could well be the result of a high reproduction by a limited numbers of successful dispersers. We now address this point more clearly in the discussion: “Comparing these numbers nevertheless requires caution, since the high nematode abundances inside ‘air only’ boxes are undoubtedly a result of the successful reproduction of dispersers on these algae; indeed, L. marina may produce several hundred progeny per gravid female [35,36]. Hence, it is impossible to estimate the number of nematodes that dispersed through air.”

We also agree that environmental conditions may have differed between treatments, but we doubt whether such differences would have been very pronounced. The “air only” and “air + seawater” boxes, for instance, were positioned at less than 20 m from eachother, so it is very doubtful whether temperature, wind etc… would have differed between them. The main environmental difference here would have been the one we wanted to create, i.e. that between inundation vs no inundation. With respect to potential differences between gauzed and open boxes: we cannot exclude differences in moisture, although in the past – when working with different mesh sizes – we found that this was mainly a problem when using gauze with smaller mesh sizes. Differences in light intensity reaching the algae would also occur, but given that both boxes with and without gauze were placed under a cover, we expect that the cover had a larger shadowing effect than the gauze. Still, we address the possibility of environmental differences between different treatments, and their potential consequences, in the discussion now: “Still, it is remarkable that dispersers did not establish similarly high population abundances in the ‘air + seawater’ boxes, because these could also be reached by vectors. One potential explanation is that flooding events may have washed away a large proportion of recently settled nematodes in these ‘air + seawater’ boxes [21,46]. Alternatively, a larger volume of water retained in these boxes may have offered a less favorable environment for reproduction of colonizing nematodes (Moens, pers obs). Finally, although the horizontal distance between the ‘air only’ and the ‘air + seawater’ boxes was small (< 20 m), and the prevailing wind direction went from the ‘air only’ to the ‘air + seawater’ boxes, we cannot completely rule out the possibility that vectors visited the latter boxes less frequently, for instance because they were placed amidst a less diverse vegetation and/or retained more water.”

Ln 181 – “the average amounts of nematodes”

Here and elsewhere in the text and the vertical axes of the figures, the authors should use “number” rather than “amounts” because nematodes are countable objects, not measured as a non-countable mass or volume. The figure legends use correct terminology.

Response: Thank you for pointing this out. We have replaced the word “amount” by “number” or “abundance” throughout the ms.

Ln 190 – Can the authors comment on how the season of each experiment (September vs April) may have affected the results? 

Response: Whilst it is a perfectly relevant question, I fear we can only speculate on the answer. There were differences in weather conditions, differences in vegetation (e.g. number of flowering plants) and many more. We have no systematic or quantitative data on any of these differences. Since we do not make any direct comparisons between the first (autumn) and second (spring) experiment, we prefer not to speculate on whether and how seasonal differences may affect the importance of aerial transport.

Ln 193 – “Four open boxes with rehydrated defaunated algae were mounted onto 3-m high sticks and placed ca. 1.75 m above the maximal tide level, while four others were positioned closer to the seaward edge of the marsh at ca. 0.27 m below the average tidal level.”

 (1) Relative to the locations of the 2 sets of boxes, where was most of the naturally occurring macroalgal wrack located?

Response: Although macroalgal wrack was present throughout the study area, albeit in a very scattered way and in low quantities (no piled-up material), the main macroalgae that could undoubtedly have served as a potential source of nematodes were living macroalgae covering a nearby breakwater (at ca. 50 m away from both treatments). Earlier research at this location has already demonstrated the abundant occurrence of L. marina on the algae on this breaker, as well as dispersal from the breaker source population to nearby patches of macroalgal wrack (Derycke et al. 2007). We have added the following sentence to the M&M section to address this concern: “Algal stands (mainly Fucus sp.) occur on a few deposits of stones mainly at the basis of the dyke and on an isolated breaker near (ca. 50 m distance from) the downstream (western) border of the marsh, whereas algal wrack is irregularly deposited in small patches along the edges of the marsh or inside small creeks and gullies and amidst vegetation.”

 (2) Note that classic experiments on airborne dispersal of small aquatic organisms has been studied by placing sampling units at different heights on the same pole (e.g., Maguire 1963 Ecological Monographs, Maguire 1971 Annual Review of Ecology and Systematics). Can the authors explain why they decided to introduce a spatial variable into their experiment by placing the “air” treatment boxes above the high tide line and the “air + water” treatment boxes lower in the tidal zone? Would it have been possible to place boxes for both treatments on the same sticks, with the “air” treatment boxes higher on the stick than the “air + water” treatment boxes? That would have allowed a nice paired t-test design and avoided possible spatial difference complications briefly mentioned by the authors.

Response: Starting with the last part of the question: we guess it would have been possible to work with boxes on poles of different height. Different boxes attached to one pole at different heights were bound to fail, because it would mean that the boxes had to be secured to the side of the poles instead of screwed tightly on top of them. Given the strong hydrodynamic action at this location, we suspect that this would have lead to loss of boxes. Also, the height of the poles was a bit of a compromise already to make the setup feasible and safe; the longer the poles, the greater the chance that they would have been pushed over by the tides. More importanty, though, we considered the potential effect of the different ‘location’ where air only and air + seawater were positioned subordinate to the different chance and frequency of receiving visits from vectors and the different identities of vectors likely to visit boxes that differed in their distance from the ground by more than 1 m. Indeed, different boxes at the same location but at a considerably different height are likely to receive different visitors and different numbers of visits. Here, we made use of a small “cliff” in the marsh, which leads to a rapid drop in height of the sediment. This allowed us to position the boxes at the same height above sediment/vegetation for both the air only and air + seawater boxes, while being able to have these two types as close to eachother as roughly 10 – 20 m, which should have greatly limited the environmental differences as well as the types of visitors between the treatments. We have now better explained this rationale in M&M. 

For more information on the microtopography of the Paulina salt marsh, the sleeves (bumps) and basins, cf. Temmerman et al. (2005), Journal of Geophysical Research. The air only and air + seawater boxes were positioned just at the border between middle tidal flat/lower tidal flat (see fig.1C in the Temmerman et al. paper). 

 (3) The experimental design is presented as “air” vs “air + water” effects on nematode dispersal. It could just as easily be described as an experiment measuring “high marsh” vs “low marsh” effects on nematode dispersal or “diverse vegetation” vs “monospecific Spartina vegetation” effects on nematode dispersal. In fact, Fig. 4 labels the 2 treatments not as “air” vs “air + water,” but as “high” and “low”! The authors need to address this problem in more detail.

Response: As stated above, the microtopography of the marsh consists of sleeves, basins, small cliffs etc… (Temmerman et al. 2005). We put our boxes at the border of the middle and the lower marsh at a distance of no more than 10 - 20 m, which was possible because of the presence of a cliff here. Admittedly, the vegetation in the lower marsh is less diverse and consists almost entirely of Spartina anglica/townsendii, whereas in the middle marsh, more plant species are present amidst the Spartina vegetation. However, given the very small distance between “air only” and “air + seawater” treatments, we consider it unlikely that mobile vectors would not have been able to move in between the two types of boxes. We have now explained this strategy in the M&M section as follows:

“We used the presence of a small cliff, abruptly ‘separating’ the middle from the lower marsh, to position boxes with defaunated algae which could only become colonized by nematodes through air, and boxes which could also become flooded at high tide at a distance of < 20 m, thereby greatly limiting the possible confounding effect of a different environment (with a different vegetation, candidate vectors etc…).”

 (4) Can the authors provide data on insect abundance and wind direction? If insects and wind are vectors, then they would strongly affect nematode dispersal into boxes at the 2 different locations.

Response: Unfortunately, we do not have data on insect diversity and/or abundance at the times and site of our study. Data on wind speed and direction are available. During the first experiment, the prevailing wind direction was north to northeast on four out of seven days and southwest on the remaining three days. During the second experiment, the prevailing wind direction was south, implying that the wind predominantly blew from the ‘air only’ to the ‘air + seawater’ boxes, hence increasing the likelihood that insects that visited the ‘air only’ boxes would also be able to visit the ‘air + seawater’ boxes. Information on temperature, wind speed and wind direction during the two field experiments has now been added to the ms.

Ln 204 – “We sampled all boxes after 14 days of exposure in the field.”

See second comment above for Ln 144 re: length of exposure period compared to length of nematode life cycle, as well as temperature and moisture differences between the treatment boxes.

Response: Comment addressed at first mention higher up.

Ln 228 – “We randomly collected 17 complete thalli of the brown alga Fucus vesiculosus at the edge of the Paulina salt marsh”

(1) Please describe the randomization procedure.

(2) Which edge of the marsh, upper or lower?

(3) Were these samples taken from live algae or from wrack?

Response: We replaced “randomly” by “haphazardly”, because there was not really a randomization procedure involved while gathering the algae. We also clarified that we collected fresh, live algae, growing near the upper edge of the marsh.

Ln 231-236 – “Each structure was then thoroughly rinsed with tap water to remove the nematodes present on the outside of the structures, before incubating it on sloppy marine agar medium (0.7% agar, 10/1 B/N, salinity of 15, see above) [47] at 20 °C in the dark. The water containing nematodes from the outer surface of each of these structures was collected separately over a 38-μm sieve and preserved in DESS [50] prior to counting.”

(1) Confusing. I do not quite understand what the authors did here. I see no mention of “inside” the structures.

(2) Please define “DESS.”

Response: Thank you for pointing this out. We have rephrased the pertinent methods section to make it clearer:

“Each structure was thoroughly rinsed with tap water; the tap water causes an osmotic shock which temporarily ‘sedates’ the nematodes, facilitating their removal from surfaces [56]. We collected the tap water and sieved it over a 38-µm sieve, then preserved it in DESS (dimethyl sulphoxide, disodium EDTA [57]) prior to counting. DESS not only preserves nematode morphology but also DNA, allowing later identification of cryptic nematode species using qPCR. After nematodes had thus been removed from their surface, the algal structures were incubated on sloppy marine agar medium (0.7% agar, 10/1 B/N, salinity of 15, see above) [53] at 20 °C in the dark, allowing nematodes from inside the receptacula and floating bladders to move out and colonize the agar.

Ln 281 – “Nematodes were present inside floating bladders and receptacula, although statistically not more than on the outside of the respective structures (Fig 5).”

(1) Is there any reason to think that the insides would have more than the outsides? That was not one of the research questions the authors set out to answer.

Response: Thanks for pointing this out. For us, it was interesting to investigate whether nematodes were more abundant on the outside than the inside of the structure, because inside they are more protected against water currents. We have rephrased the original sentence to: 

“Nematodes were present inside floating bladders and receptacula, where they are likely more sheltered against water currents, as well as on the outside of the respective structures (Fig 5).”

(2) See misspelling on horizontal axis of Fig. 5 “outiside” should be “outside”)

Response: Thanks for pointing this out. It has been corrected.

Ln 291 – “The active dispersal capacity of marine nematodes is limited because of their small body size, limited swimming ability and lack of larval stages that can disperse (reviewed in [13]). Hence, dispersal over both short and long distances is generally believed to be largely passive [10,13] and to result from transport through the water column [11]. Alternative dispersal mechanisms in marine nematodes have received only limited attention [16,17].”

Yes, nematode dispersal is largely passive. But that does not mean that such dispersal is rare or unimportant. Please see comments above for Ln 46. Bedload transport (rather than water column transport per se) has been shown repeatedly to be an important dispersal mechanism for marine nematodes, and it is strongly controlled by wind-generated hydrodynamic forces. These dispersal rates have been shown to vary according to nematode feeding type. Some feeding types live closer to the bed surface and are more easily eroded and transported. In fact, epigrowth-feeders have by far the highest absolute, relative, and bulk dispersal rates.

Overall, nematodes have very high absolute dispersal rates, as would be expected given their abundance in marine sediment. However, when normalized to a per capita basis, they do not disperse at rates as high as meiofauna with active emergence behavior, such as copepods. (See Discussion in Commito et al. 2019 Diversity for comparison of per capita bedload transport rates of nematodes vs copepods, with data from a variety of locations.) Thus, it is important to investigate other forms of dispersal for nematodes, including airborne and rafting modes of nematode dispersal.

Response: Thank you for this comment. We have now rephrased the introductory § of the discussion section to:

“The active dispersal capacity of marine nematodes is limited because of their small body size, limited swimming ability and lack of larval stages that can disperse (reviewed in [13]). Nevertheless, overall dispersal rates of nematodes can be high, mainly as a consequence of bedload transport [7-10]. However, on a per capita basis, they do not disperse at rates as high as meiofauna with active emergence behavior, such as copepods [10]. It is therefore important to investigate other forms of dispersal for nematodes, including airborne transport and rafting.”

Ln 325 – “Hitchhiking through air on invertebrate vectors provides a more direct dispersal mechanism for nematodes associated with deposits of algal wrack, and may have many benefits above random dispersal through water currents, because macroalgal patches are sparsely distributed and often ephemeral habitats, and because nematodes lack good active swimming capacities, whilst at least some of their candidate vectors can efficiently move between such patches.”

(1) For nematodes that can live in macroalgal patches and in sediments, the nematodes may move by bedload transport directly from sediment onto macroalgal patches. They may also “hop-scotch” from a macroalgal patch to the sediment and then from the sediment to another macroalgal patch. The authors need to address this point.

Response: We do not fully agree with this point. It is correct that bedload transport may also facilitate movement between sediment and macroalgae, and between macroalgal patches. However, bedload transport seems a relatively non-selective (in terms of habitat choice) mode of dispersal, transporting meiofauna from one location to another, but with limited possibility for this meiofauna to choose a new location. The point we wanted to make at this position in the discussion is that vector-mediated transport is likely more specific and therefore – at least in the short term – efficient, because the nematodes associate with vectors that are likely to visit habitat patches that are highly suitable for the nematodes. L. marina, for instance, is only very occasionally found in marsh sediment, even close to deposited algal wrack, which suggests that a dispersal mode which involves hop-scotching between suitable and unsuitable habitat may not be an efficient strategy for this nematode. See also Arroyo, N. L., Aarnio, K., & Bonsdorff, E. (2006). Drifting algae as a means of re-colonizing defaunated sediments in the Baltic Sea. A short-term microcosm study. Hydrobiologia, 554, 83: “Contrary to our predictions, nematode and harpacticoid species inhabiting the drifting algae were not driven to sediment re-colonization but remained in the algae”.

We therefore prefer not to re-iterate bedload transport here, since we are now already pointing at its importance at multiple locations in introduction and discussion. 

(2) Change “above random dispersal” to “beyond random dispersal” or “in addition to random dispersal”

Response: We have rephrased this to “beyond random dispersal” as suggested by the reviewer.

(3) Water currents certainly have a random component due to turbulence. But water currents typically have strong directional component due to the ebb and flow of the tides as well as to wind direction. These water currents move meiofauna, including nematodes, in predictable (not random) directions (e.g., Fegley 1988 JEMBE, DePatra and Levin 1989 JEMBE, Commito et al. 2019 Diversity).

Response: We have removed the comparison with water currents at this point, as it only distracts from the point we wanted to make about the non-random nature of the vector transport. Also, while we agree that water currents are not random in terms of the direction of the dispersal they facilitate, we feel that they do not provide a similarly efficient means of habitat-selective transport.

Ln 335 – Please define “entomopathogenicity” and “phoresy,” which I do not think most PLOS ONE readers will know.

Response: The terms entomopathogenicity and phoresy have been replaced by parasitism on insects and vector-mediated dispersal, respectively.

Ln 339-355 – I am glad that the authors presented several possible explanations why the “air” boxes had higher numbers of nematodes than did the “air + water” boxes. The 2 treatments were established in 2 very different habitats as well as at 2 different distances from the marine source of nematodes. So the experimental results are somewhat compromised by the experimental design, as mentioned above in my comments for Ln 193. However, the results do point the way to future research.

Response: We have modified the pertinent text fragment to “Finally, although the horizontal distance between the ‘air only’ and the ‘air + seawater’ boxes was small (< 20 m), and the prevailing wind direction went from the ‘air only’ to the ‘air + seawater’ boxes, we cannot completely rule out the possibility that vectors visited the latter boxes less frequently, for instance because they were placed amidst a less diverse vegetation and/or retained more water.” 

Ln 373 – “Nematodes may search for protection on and inside the latter structures”

Perhaps nematodes exhibit this type of searching behavior, but do the authors believe it is likely that nematodes actively search for such protection? Is there any evidence for it in the literature?

Response: We rephrased this to “Nematodes may benefit from protection against water currents and predation inside the latter structures”, as it was not our intention to discuss whether or not the nematodes actively search for this protection. Nor is the surface of these structures a plausible location where nematodes are protected against currents or predation (see De Meester et al. 2018 – Mar Ecol Prog Ser). The suggestion that nematodes may seek protection against predation by moving inside or in between macrophyte structures was also made by Walters et al. for nematodes inhabiting stems of the cordgrass Spartina alterniflora: Walters K, Jones E, Etherington L. Experimental studies of predation on metazoans inhabiting Spartina alterniflora stems. J Exp Mar Bio Ecol. 1996 Feb 29; 195: 251-265.

Ln 386 – “Further dedicated research will have to reveal which specific associations exist, and whether other, more sediment-bound vectors that actively move in between suitable habitat spots (such as crabs) could also serve the same role, transporting nematodes from and to suitable habitat patches.”

See comments above for Ln 46, 291, and 325. Bedload transport is an example of a “more sediment-bound vector.”

Response: Comment accepted. We have rephrased this sentence as follows:

“Further dedicated research will have to reveal which specific associations exist, whether other vectors that actively move in between suitable habitat spots (such as crabs) could also serve the same role, transporting nematodes from and to suitable habitat patches, and what is the quantitative importance of such vector-mediated dispersal relative to dispersal through bedload transport.”

Ln 396 – “This may explain existing evidence for “long-distance dispersal events” in Litoditis marina and perhaps other marine nematodes, and may therefore provide one answer to the meiofauna paradox: meiofauna lack larval dispersal stages or active larger-scale dispersal ability, yet quite many species have very extensive geographic distributions. As such, the results of the present study can be important for our understanding of both large-scale geographic distribution patterns and of the small-scale colonization dynamics of habitat patches by marine nematodes.”

I certainly agree that drifting macroalgae can be a long-distance dispersal mechanism for nematodes. However, as mentioned above in my comments for Ln 62, I do not believe that a meiofaunal paradox exists. A species does not need long-distance dispersal events in order to have an extensive geographic distribution. Daily short-distance bedload dispersal may carry organisms just as far as infrequent long-distance dispersal events.

Response: We have removed this sentence from the conclusions.

Reviewer #4: Review of Colonization of macroalgal deposits by estuarine nematodes through air and potential for rafting inside algal structures. PONE-D-20-18326.

This is an interesting and novel piece of fundamental ecological research and I recommend it for publication in PLOS One after minor revisions. Whilst there are no major flaws that I have identified (although some confounding in experiment 2: see comments below), the text is a little unclear in several places and requires some clarification to enhance the message. The discussion is also rather repetitious and could be shortened and tightened.

Response: Thank you for your interest. We have tried to improve the message of the manuscript, making it more comprehensive and clear. We have also scrutinized the discussion and conclusions sections, reducing their respective lengths by 24.3 and 20% (based on word count), mainly by removing repetitive sections.

Minor comments:

Abstract: sentence from 30-33 is rather long-winded and it makes it difficult to read. I suggest to shorten this perhaps start “We also demonstrate for the first time….” And remove “and even quite efficiently”.

Response: Comment has been adopted.

Line 60: what is meant by “subsequent phases”? do you mean recruitment? Please specify. 

Response: We have removed this sentence as it was rather redundant after the previous one.

Line 70 onwards: after first mention, Litoditis marina can be referred to as L. marina.

Response: Done, except when a sentence starts with this name.

Line 94: grammatically the “both” should come occur after “nematode dispersal” as in “we investigated nematode dispersal both through air…”

Response: This sentence has been rephrased and the grammatical error pointed out by the reviewer removed.

Line 96 to 103 seems out of place in an introduction and would perhaps be better in the methods since it is describing what was done.

Response: We have removed this paragraph from the introduction. It has not been included as such in the methods, since its content is already sufficiently covered in several sections of the methods.

Please add a hypothesis or hypotheses to the end of the introduction.

Response: While we understand this comment, in the sense that the formulation of clear research hypotheses usually renders a ms more attractive to the reader, this was an exploratory study. There are underlying hypotheses, for instance that nematodes can utilize invertebrate vectors to disperse in between habitat patches, and that nematodes can hide inside specific structures of algae, which may be an important mechanism explaining long-distance dispersal events of marine nematodes. However, we did not/could not test whether long-distance dispersal of nematodes happens in this way, nor could we check the actual association of nematodes with invertebrate vectors. The present study comes one step before such specific hypothesis-driven studies, by demonstrating the plausibility of these phenomena. Therefore, we prefer to write the aims/goals of the study in a more descriptive way, rather than by formulating hypotheses.

Line 123 and elsewhere: by “water” is “seawater” meant or rain? Please be specific. 

Response: This has been clarified throughout the ms.

Line 137: the mesh is 200um so which visiting animals is this allowing entrance for? I thought that the mesh was the allow wind dispersal but prevent vector dispersal on insects? This is a little unclear, please clarify.

Response: The reviewer’s interpretation is correct. However, vectors could still land on the boxes covered with gauze, and hitchhiking nematodes could then potentially become detached from their vectors and enter through the gauze. We have rephrased this sentence to clarify this point: 

“boxes covered with a gauze with a mesh size of 200 µm , preventing entry of insects, spiders and other potential vectoring fauna, but allowing entry of nematodes transported by wind or dropping off from vectors landing on top of the gauze.”

Line 151: metal piles should be metal poles? Also in figure 1 and elsewhere. 

Response: Comment adopted.

Methods: Are the boxes water-tight? Are they lined with something to make them hold water?

Response: Line 150, added now the fact they had waterproof silicon lining.

Line 156: write the whole word for diameter.

Response: Comment adopted.

Why were the nematodes dead in the first experiment and so few in abundance? Some further exploration of this in the discussion would be good. Was it due to the methods, did the boxes get too hot or dry out?

Response: We suspect the algae inside the boxes partly dried out, leading to a large increase in the salinity of the remaining water. We have added the following explanation to the text: 

“The fact that only dead nematodes were obtained from the boxes in the first experiment indicates that environmental conditions in the boxes rapidly turned unfavourable, possibly through a pronounced increase in the salinity of the ASW following evaporation.”

Line 194 and throughout: “Sticks” or “poles”? Please be consistent in some cases these are also called “piles”. Also what were these poles made out of?

Response: Thanks for pointing this out. We have now consistently used the word “poles”. They were made from metal, see text (“metal poles”) and figure description.

Line 199: Some more detail is required on how much water these “low” boxes received. How long were they immersed for? Did the whole box become covered?

Response: The boxes received water every tidal cycle, meaning that water could have been flushed in and out, but as tides are not consistent in time, the amount is impossible to assess. The boxes stood 0.27 m below average tide level, at the border of the higher and middle tidal flat. This info has been added in the ms. 

Line 200: treatments are called “air only” and “air + water” but are called “low” and “high” in figure 4. Please be consistent.

Response: This has been modified in figure 4 to improve consistency. We now use “air only” and “air + seawater” throughout.

~line 216: how many nematodes or what % was accurately sampled?

Response: As can be seen from the accompanying figure, numbers of nematodes inside the “air only” boxes amounted to several thousands, whereas the corresponding numbers inside “air + seawater” boxes were more than an order of magnitude lower. Hence, 100 nematodes represented just a little more than 1% of the nematodes inside the “air only” boxes, but roughly half the nematodes inside the “air+seawater” boxes. Please note that the purpose of this identification was merely to assess the presence of Litoditis marina, not to estimate its abundance. 

Line 223: why log- transformed?

Response: Data had to be log-transformed to fit the assumptions of norma distribution and homoscedasticity.

~Line 233: Bit unclear how the inside versus outside of the algal structures were sampled. Rinsed to remove outside, but then were the structures cut to release the nematodes from within? Is there a protocol to refer to that proves that this technique works? Is it possible that some nematodes hold on very strong and are still present on outside even after rinsing? Please provide some extra evidence here.

Response: Rinsing the surface of the algae using a jet of tap water causes an osmotic shock which causes the nematodes to stretch out and become detached from any surfaces they are adhering to at the time; the same procedure is commonly used to obtain large quantities of living nematodes from estuarine/marine sediments (e.g. Derycke et al. 2007, Moens & Vincx 1998, Guden et al. 2018; Beninger & Moens, 2018). Whether this procedure is 100% effective in removing surface-associated nematodes has, to our knowledge, never been rigorously tested. Still, De Meester et al. (2018) observed that more than 60% of L. marina living on the same Fucus substrate were washed off the algae in even relatively mild currents of seawater (20 cm s-1 during 30 min), so without the osmotic shock and the more vigorous physical rinsing applied here, suggesting that the vast majority of nematodes would definitely be removed by the procedure applied here. 

We did not cut the bladders or receptacles to facilitate the ‘release’ of nematodes from inside these structures. Microbial growth in the plates rapidly caused a decay of these structures, allowing nematodes to easily move out. But since they are able to get in the intact structures, they most probably can also find their way out.

Line 241: the “dedicated qPCR protocol” needs some explanation here, it is ok to be brief, but simply referring to another paper for an entire method is too vague. Also how many samples were analysed in this way? 

Response: Agreed. The following information has been added: “for identification to species level using a quantitative real-time PCR of the ribosomal ITS region; this PCR was performed with four different primers, each of which matches specifically and exclusively with one of the four cryptic species of L. marina known from our study area.”

Line 251: “resp.?” please write in full.

Response: Comment adopted. 

Line 253: what programme was used for the analysis and which packages with references please. 

Response: We added the following info to the text:

“…were compared using a linear mixed model, implemented in the R software program (version 4.0.3, R Core Team 2020) with packages nlme version 3.1 [68] and lme4 version 1.1 [69], with algal structure and inside vs. outside as fixed factors.”

Line 266: Suggest using “greater amount” instead of “higher amount” since this confuses with the height on the shore.

Response: We have replaced “higher amount” by “larger number” when referring to the abundances of nematodes (see also comment of ref. 3 with respect to the use of ‘number’ instead of ‘amount’).

Line 284: “the few” is too vague. How many samples and what proportion?

Response: We have specified the number of nematodes we identified in this way between brackets.

Line 310: wind transport is discussed here but the mesh treatments in the first experiment had no nematodes thereby it was likely to be vector based transport? Bit confusing. I suggest moving the sentence in 315 upwards and moving or removing the sentence in 313. The flow of this discussion on this point is confusing and also there is repetition with the introduction so not all of this text is needed. Could remove from 310 to 315.

Response: Agreed. We have removed the first sentence, which we think caused the confusion. We also modified the following sentences, mainly in response to comments by reviewer 3.

Line 328-331: Does this explain why there were so few nematodes in the water + air treatments? This section could be related more clearly to experiment 2.

Response: See answer to the next question/comment for discussion on possible reasons for the low nematode abundances in the ‘air + seawater’ boxes.

Discussion: Why do the lower boxes have so few nematodes, could it be that the seawater is flushing them back out of the boxes as the water ebbs in and out? I think a sentence or two is needed in the discussion to address this explicitly. I see it occurs in lines 342 onwards, but think it needs to be earlier, so suggest to move this up to around line 328 and shorten the whole section.

Response: We have added the following explanation in the discussion: “Still, it is remarkable that dispersers did not establish similarly high population abundances in the ‘air + seawater’ boxes, because these could also be reached by vectors. One potential explanation is that flooding events may have washed away a large proportion of recently settled nematodes in these ‘air + seawater’ boxes [21,46]. Alternatively, a larger volume of water retained in these boxes may have offered a less favorable environment for reproduction of colonizing nematodes (Moens, pers obs). Finally, although the horizontal distance between the ‘air only’ and the ‘air + seawater’ boxes was small (< 20 m), and the prevailing wind direction went from the ‘air only’ to the ‘air + seawater’ boxes, we cannot completely rule out the possibility that vectors visited the latter boxes less frequently, for instance because they were placed amidst a less diverse vegetation and/or retained more water.”

Line 332-338: this seems repetitious too, please check for repetition with the intro and preceding discussion and try to reduce.

Response: Although we cannot fully exclude some repetition here, the discussion has been scrutinized and shortened by removing redundant (repetitive) info.

Line 340: greater instead of higher. This occurs a few times, grammatically it is clearer to use “greater”.

Response: Depending on the context, we replaced “higher” by “larger” or “greater”.

The treatments in the second experiment which aim to separate “water + air” with “air only” are confounded with height on the shore which also includes temperature differences as well as humidity and moisture and possibly fauna. One way to separate this could have been to wet the “high boxes” with fresh seawater each day for the same amount of time and volume that the “low boxes” received. Is there evidence that the insect or arthropod vectors visit the lower shore at all? The manuscript states that they “probably” do. Can this be supported by literature or at least acknowledged as an unknown? This confounding is partly acknowledged around lines 347 but more is required please.

Response: Essentially the same comment was raised by reviewer 3. See our responses to his/her comments on this issue. In summary, we have provided the following info in the revised ms:

In the M&M section:

 “We used the presence of a small cliff, abruptly ‘separating’ the middle from the lower marsh, to position boxes with defaunated algae which could only become colonized by nematodes through air, and boxes which could also become flooded at high tide at a distance of < 20 m, thereby greatly limiting the possible confounding effect of a different environment (with a different vegetation, candidate vectors 2000 etc…).”

In the discussion section:

“Finally, although the horizontal distance between boxes higher up and lower down in the marsh was small (< 20 m), we cannot completely exclude that potential vectors visited the boxes that were positioned lower less frequently, since the higher boxes were placed amidst a more diverse vegetation (including species such as Aster tripolium, Atriplex portulacoides and Limonium vulgare in addition to the dominant Spartina x townsendii) than the lower boxes, which were positioned in almost monospecific S. x townsendii vegetation.”

Line 378: please remove the etc…..

Response: Comment adopted.

Line 378: “robust” in what way?

Response: Indeed, this was a very subjective interpretation. We have changed the sentence to “Similarly, floating bladders may provide good shelter for nematodes.”, thus also removing the word “robust”. 

Line 398-399: “quite many species” is vague, please replace.

Response: We have removed this sentence and conclude instead with the following final sentence, which leaves it more open whether or not the observed mechanisms of dispersal are more commonly used by a variety of nematode species: “Further research will also have to elucidate whether this mechanism is restricted to marine Rhabditidae, a nematode family where many terrestrial representatives can hitchhike on insects using their dauer larval stage, or whether the phenomenon is more broadly used by nematodes from intertidal environments.” 

All figures: use capital letters at start of axis titles.

Response: Comment adopted.

Figure 2: do you have or require permission to use this image? 

Response: Figure has been replaced by (part of) a drawing which we have also used in a previous paper from our team in PLoS One (Guden et al. 2018 - https://doi.org/10.1371/journal.pone.0204750.g001).

Figure 3: y-axis title is a bit too long, perhaps “Average abundance of nematodes”, the per box part can be in the figure legend.

Response: Comment adopted.

---

## [Editor Report · Decision Letter 1]

26 Jan 2021

Colonization of macroalgal deposits by estuarine nematodes through air and potential for rafting inside algal structures

PONE-D-20-18326R1

Dear Dr. Moens,

We’re pleased to inform you that your manuscript has been judged scientifically suitable for publication and will be formally accepted for publication once it meets all outstanding technical requirements.

Kind regards,

Maura (Gee) Geraldine Chapman, PhD DSc

Academic Editor

PLOS ONE
---

## [Editor Report · Acceptance letter]

5 Apr 2021

PONE-D-20-18326R1 

Colonization of macroalgal deposits by estuarine nematodes through air and potential for rafting inside algal structures 

Dear Dr. Moens:

I'm pleased to inform you that your manuscript has been deemed suitable for publication in PLOS ONE. Congratulations! Your manuscript is now with our production department. 

Kind regards, 

on behalf of

Professor Maura (Gee) Geraldine Chapman 

Academic Editor

PLOS ONE